# Structural characterization of tin in toothpaste by dynamic nuclear polarization enhanced $^{119}$Sn solid-state NMR spectroscopy

Rick W. Dorn[1,2], Scott L. Carnahan [1,2], Chi-yuan Cheng[3], Long Pan[3], Zhigang Hao [3] ✉ & Aaron J. Rossini [1,2] ✉

Stannous fluoride ($SnF_2$) is an effective fluoride source and antimicrobial agent that is widely used in commercial toothpaste formulations. The antimicrobial activity of $SnF_2$ is partly attributed to the presence of Sn(II) ions. However, it is challenging to directly determine the Sn speciation and oxidation state within commercially available toothpaste products due to the low weight loading of $SnF_2$ (0.454 wt% $SnF_2$, 0.34 wt% Sn) and the amorphous, semi-solid nature of the toothpaste. Here, we show that dynamic nuclear polarization (DNP) enables $^{119}$Sn solid-state NMR experiments that can probe the Sn speciation within commercially available toothpaste. Solid-state NMR experiments on $SnF_2$ and $SnF_4$ show that $^{19}$F isotropic chemical shift and $^{119}$Sn chemical shift anisotropy (CSA) are highly sensitive to the Sn oxidation state. DNP-enhanced $^{119}$Sn magic-angle turning (MAT) 2D NMR spectra of toothpastes resolve Sn(II) and Sn(IV) by their $^{119}$Sn chemical shift tensor parameters. Fits of DNP-enhanced 1D $^1$H → $^{119}$Sn solid-state NMR spectra allow the populations of Sn(II) and Sn(IV) within the toothpastes to be estimated. This analysis reveals that three of the four commercially available toothpastes contained at least 80% Sn(II), whereas one of the toothpaste contained a significantly higher amount of Sn(IV).

Oral health is highly important for a person's overall physical health and well-being[1–4]. Toothpaste is the most commonly used dental hygiene product to maintain oral health by preventing and protecting against oral diseases, such as caries (i.e., cavities/tooth decay) and gingivitis (i.e., gum disease)[5]. Within commercial toothpastes, fluoride is the most effective active ingredient to prevent caries/cavities[1,5–10]. Stannous fluoride ($SnF_2$), one of three fluoride sources recognized by the United State Food and Drug Administration, has been used since the 1950's as an effective way to deliver fluoride ions to tooth enamel and dentin[11–14]. However, early toothpastes exhibited challenges associated with the stability of $SnF_2$ and its compatibility with other ingredients[14–17]. Fortunately, advances in formulation technologies over the last couple decades have resulted in toothpastes with reportedly stable $SnF_2$[14,17–21].

The use of $SnF_2$ as a fluoride source is highly appealing because $SnF_2$ has been shown to be an effective antimicrobial agent; antimicrobial agents help reduce gingivitis and plaque formation[15,21–30]. The antimicrobial properties associated with $SnF_2$ in toothpaste is thought to arise from the presence of Sn cations with an oxidation state of +2 [Sn(II)][21,23,31–33]. Therefore, it is important that the majority of Sn within commercially available toothpaste maintains the +2 oxidation state[21,23,31–33]. However, Sn(II) cations are generally unstable and will readily oxidize to the more stable Sn(IV) cations upon air exposure. Commercial toothpaste formulations have various additives designed to stabilize Sn(II) compounds within the formulation. But, it is challenging to directly determine the Sn speciation and oxidation state within commercially available toothpaste due to the low weight loading of $SnF_2$ (0.454 wt%, 0.34 wt% Sn) and the amorphous semi-solid

[1]US Department of Energy Ames National Laboratory, Ames, IA 50011, USA. [2]Department of Chemistry, Iowa State University, Ames, IA 50011, USA. [3]Colgate-Palmolive Company, Piscataway, NJ 08855, USA. ✉e-mail: zhigang_hao@colpal.com; arossini@iastate.edu

nature of the toothpaste. In 2019, Myers and co-workers used Sn K-edge X-ray Absorption Near Edge Spectroscopy (XANES) to determine that Colgate Total[SF] contains approximately 85% of its tin in the Sn(II) oxidation state[17]. More recently, Desmau and co-workers probed the Sn oxidation state within commercially available toothpastes via Sn K-edge XAS[34]. In both reports, the relative populations of Sn(II) and Sn(IV) species were estimated by fitting the experimental XAS spectra with XAS spectra of Sn(II) and Sn(IV) standards. However, XAS provides only a partial picture of chemical structure, and energy differences of the Sn(II) and Sn(IV) Sn K-edge spectral features are relatively small (~2.5 eV) and overlapping.

Magic-angle spinning (MAS) solid-state NMR spectroscopy is a powerful technique to determine structure within crystalline and amorphous solids. Sn possess three NMR active isotopes ([115]Sn, [117]Sn, and [119]Sn), with [119]Sn generally being the preferred nucleus to probe because it has the largest gyromagnetic ratio (2.7 times lower than [1]H) and natural isotopic abundance (8.59%). [119]Sn exhibits a large isotropic chemical shift range that is highly dependent on the local electronic structure surrounding the Sn atom[35-45]. In addition, the magnitude of the chemical shift anisotropy (CSA) is highly dependent on the symmetry at the Sn atom; asymmetric Sn coordination environments yield large CSA and broad NMR spectra. The magnitude of CSA is often quantified with the span ($\Omega$) which is calculated from the difference of the largest and smallest principal components of the magnetic shielding tensor ($\Omega = \sigma_{33} - \sigma_{11}$) or the chemical shift tensor ($\Omega = \delta_{11} - \delta_{33}$)[46,47]. In general, Sn(IV) adopts a much more symmetric structure than its corresponding Sn(II) analogs and likely has larger differences in energy between occupied and unoccupied orbitals. For these reasons, Sn(IV) compounds tend to have smaller CSA than Sn(II) compounds[40,48,49]. For example, [119]Sn NMR spectra of Sn(IV) systems typically reveal spans of 0–200 ppm[36,40,48-55], whereas for Sn(II) compounds, the span is often between ~ 700–1000 ppm and can be upwards of ~4000 ppm[39-41,48,49,53,56-62]. We note that there have been a few reports of Sn(IV) or Sn(II) exhibiting relatively large or small spans, respectively, due to either distorted or highly symmetric coordination environments, respectively[49,62-65]. Unfortunately, conventional room temperature [119]Sn MAS NMR spectroscopy of commercial toothpaste is not practically feasible due to the low weight loading of Sn within commercial toothpastes (~ 0.34 wt% Sn). In addition, the semi-solid nature of toothpaste will likely result in molecular mobility that causes the full or partial averaging of the [119]Sn CSA at room temperature. Additionally, solution [119]Sn NMR is also hindered by the semi-solid nature of toothpaste.

Here, we apply cryogenic MAS dynamic nuclear polarization (DNP)[66-69] to enhance [119]Sn solid-state NMR signals of frozen toothpastes by one to two orders of magnitude, allowing 1D [119]Sn solid-state NMR spectra to be obtained in minutes from dilute commercial toothpaste formulations. Conventional room temperature solution and solid-state NMR experiments are performed on $SnF_2$ and $SnF_4$. In a MAS DNP experiment, microwave irradiation is used to saturate electron paramagnetic resonance transitions, resulting in the subsequent transfer of electron spin polarization from stable free radicals to the [1]H spins of the solvent matrix and/or analyte[66-68]. Toothpastes typically contain high amounts of water and/or glycerol; water/glycerol mixtures have been shown to be an ideal matrix for MAS DNP experiments[66]. We note that the use of DNP to enable the acquisition of [119]Sn solid-state NMR spectra of dilute Sn(IV) species has been previously demonstrated[55,70-75]. To enable DNP experiments the AMUPol biradical[76] was directly dissolved in the toothpaste formulation. The DNP experiments were performed at ca. 110 K on frozen toothpaste. Freezing the toothpaste is beneficial because it eliminates any molecular motion, allowing measurement of [119]Sn CSA and enables [1]H-[119]Sn cross-polarization (CP) to transfer the DNP-enhanced [1]H polarization to [119]Sn nuclei. The [119]Sn CSA is shown to be a sensitive probe of the Sn oxidation state and from fits of the 1D [119]Sn solid-state NMR spectra the

relative amounts of Sn(II) and Sn(IV) within the formulation can be estimated.

## Results and discussion

Sn(II) and Sn(IV) Fluoride—[19]F and [119]Sn Chemical Shifts and Sample Purity Analysis. We first performed room temperature [19]F and [119]Sn solid-state NMR spectroscopy on $SnF_2$ and $SnF_4$ to determine chemical shift tensor parameters of Sn in the II or IV oxidation state, respectively. $SnF_4$ features Sn in the more stable IV oxidation state, with each Sn atom residing in a symmetric octahedral environment coordinated by 6 F atoms (Fig. 1A)[77]. Consequently, no spinning sidebands are observed in the [119]Sn solid-state NMR spectrum of $SnF_4$ recorded with a 25 kHz MAS frequency, indicating $\Omega$ is less than 170 ppm (Fig. 1B, upper). Note throughout this entire manuscript, the Herzfeld-Berger convention is used to report the CSA[46,47]. $\alpha$-$SnF_2$ contains Sn in the less stable, II oxidation state and there are two unique Sn sites in asymmetric environments that are coordinated by three or five F atoms (Fig. 1A)[78]. 1D [119]Sn NMR spectra of $SnF_2$ were recorded for samples purchased from two different suppliers. The [119]Sn NMR spectrum of $SnF_2$ (supplier **b**) reveals two isotropic [119]Sn NMR signals with $\delta_{iso}$ = −948 ppm or −1023 ppm and $\Omega$ = 990 ppm or 930 ppm, respectively (skew = $\kappa$ = 1.0 in both cases; Fig. 2B, lower). The [119]Sn NMR spectrum of $SnF_2$ (supplier **b**) is consistent with prior reports and the structure determined from single-crystal X-ray diffraction[59,79]. On the other hand, the sample from supplier **a** shows additional [19]F and [119]Sn NMR signals. These additional NMR signals are attributed to Sn(IV) fluoride impurities because the observed [19]F chemical shifts, [119]Sn CSA, and correlations observed in the 2D [19]F{[119]Sn} J-HMQC spectrum are similar to those observed for $SnF_4$ (Fig. 1B and 2 and Supplementary Fig. 2). We note that the [19]F chemical shift of tin fluoride materials appear to be sensitive to the Sn oxidation state; $SnF_2$ and $SnF_4$ exhibit a $\Delta\delta_{iso}$([19]F) of 100 ppm (Fig. 2A).

Periodic plane-wave density-functional theory (DFT) calculations utilizing the gauge-including projector-augmented wave (GIPAW) method predicts that the [119]Sn isotropic shielding ($\sigma_{iso}$) and $\Omega$ of two Sn species in $SnF_2$ differ by 76 and 68 ppm, respectively, and that the most shielded (i.e., most negatively shifted) Sn site exhibits the smaller $\Omega$ (Table S1). The most shielded Sn site is coordinated by 5 F atoms. The predicted difference in $\sigma_{iso}/\delta_{iso}$ and $\Omega$ is in excellent agreement with that observed experimentally, where the $\delta_{iso}$ and $\Omega$ differ by ca. 75 and 60 ppm, respectively. Therefore, we assign the two [119]Sn NMR signals with $\delta_{iso}$ = −948 ppm ($\Omega$ = 990 ppm) or −1023 ppm ($\Omega$ = 930 ppm) to Sn coordinated by three or five F atoms, respectively. We note that while the difference in the DFT calculated $\Omega$ is in excellent agreement with that observed experimentally, the magnitude of the DFT calculated $\Omega$ is smaller (Table S1). GIPAW calculations do not account for relativistic effects, which are likely needed to accurately calculate [119]Sn CS tensors[80,81]. GIPAW calculated [19]F chemical shielding values are in reasonable agreement with experiment (Supplementary Fig. 1). The calculations predict that $SnF_2$ should have [19]F chemical shifts which are approximately 100 ppm more positive than those of $SnF_4$.

The [119]Sn NMR spectrum of $SnF_4$ shows multiple [119]Sn NMR signals that all exhibit small CSA, revealing that this $SnF_4$ sample contains impurities (Fig. 1B, upper). A 1D [19]F spin echo NMR spectrum reveals three NMR signals between ca. −120 ppm to −160 ppm (Fig. 2A); but, only two [19]F NMR signals are expected for the terminal and bridging F atoms in a 1:2 ratio, respectively (Fig. 1A). We recorded a 2D [19]F{[119]Sn} J-HMQC NMR spectrum to better probe the $SnF_4$ species from the impurities within this material (see Supplementary Material, Supplementary Fig. 1). Only the [119]Sn NMR signals at ca. −750 ppm and −785 ppm show correlations to two unique [19]F NMR signals at ca. −130 ppm and −150 ppm; the difference in the [19]F isotropic chemical shift [$\Delta\delta_{iso}$([19]F)] for the latter two sites agrees with that predicted by GIPAW DFT calculations ($\Delta\sigma_{iso}$([19]F) = 26 ppm, Supplementary Table 1). Therefore, both [119]Sn NMR signals at ca. −750 ppm and −785 ppm could

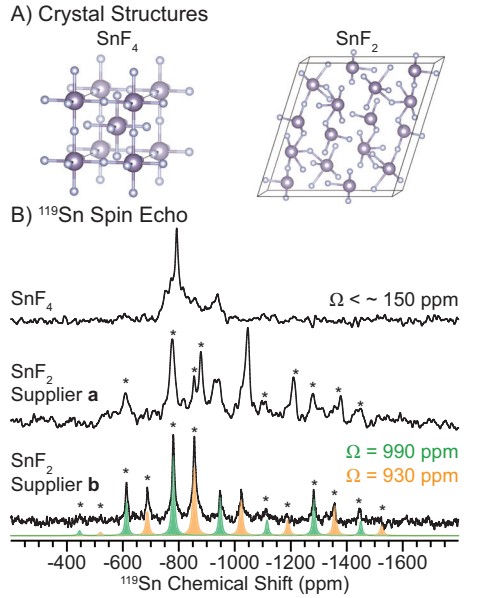

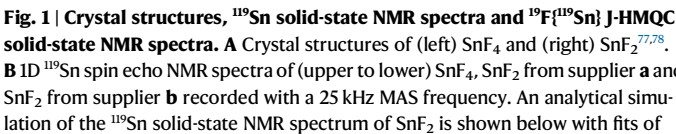

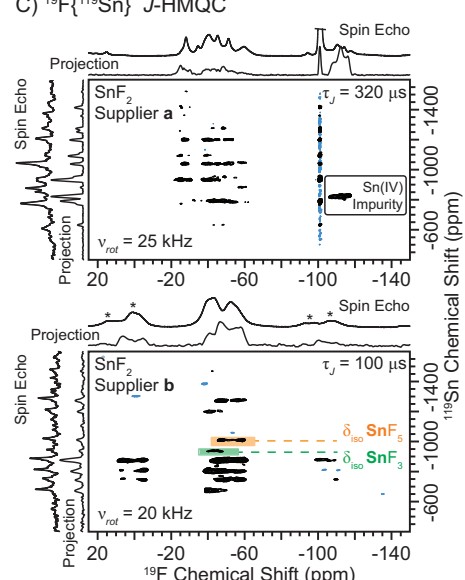

**Fig. 1 | Crystal structures, ¹¹⁹Sn solid-state NMR spectra and ¹⁹F{¹¹⁹Sn} J-HMQC solid-state NMR spectra. A** Crystal structures of (left) $SnF_4$ and (right) $SnF_2$[77,78]. **B** 1D ¹¹⁹Sn spin echo NMR spectra of (upper to lower) $SnF_4$, $SnF_2$ from supplier **a** and $SnF_2$ from supplier **b** recorded with a 25 kHz MAS frequency. An analytical simulation of the ¹¹⁹Sn solid-state NMR spectrum of $SnF_2$ is shown below with fits of sideband intensities shown for the $SnF_3$ and $SnF_5$ sites (solid green and orange peaks, respectively). Estimated and fitted chemical shift tensor spans ($\Omega$) are indicated. **C** 2D ¹⁹F{¹¹⁹Sn} J-HMQC NMR spectra of $SnF_2$ from (upper) supplier **a** and (lower) supplier **b**. Spectra were recorded with J-evolution times ($\tau_J$) of 100 µs. Asterisks (*) indicate spinning sidebands.

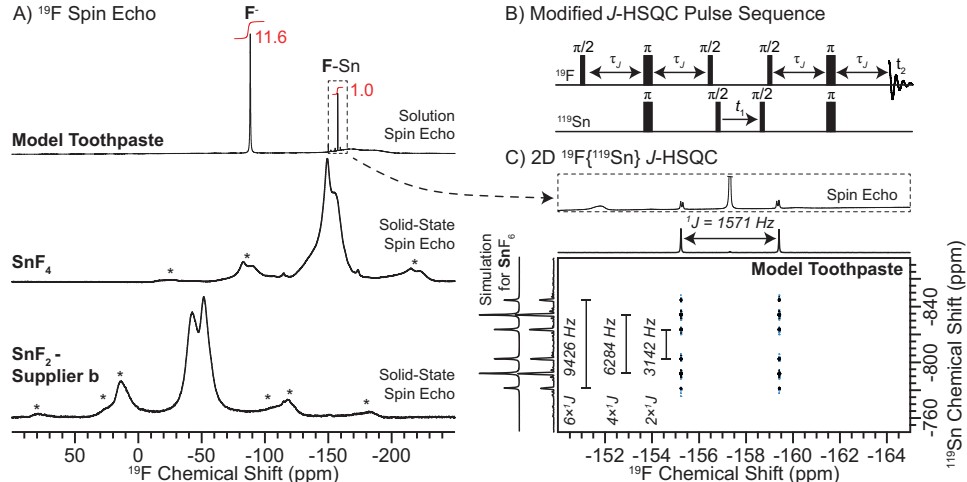

**Fig. 2 | Solution and solid-state ¹⁹F NMR spectra and solution ¹⁹F{¹¹⁹Sn} HSQC NMR spectrum. A** ¹⁹F spin echo NMR spectra of (upper to lower) the model toothpaste, $SnF_4$ and $SnF_2$ (supplier **b**) recorded in either (upper) solution or (lower two) the solid-state. Integrated peak intensities are indicated on the solution ¹⁹F NMR spectrum (red). Asterisks indicate spinning sidebands. **B** Modified ¹⁹F{¹¹⁹Sn} J-HSQC pulse sequence that allows for evolution of the ¹⁹F-¹¹⁹Sn J-coupling during the ¹¹⁹Sn $t_1$-evolution period. $\tau_J$ denotes time periods for evolution of heteronuclear J-couplings. Note the absence of the ¹⁹F π-pulse in the center of the $t_1$-evolution period. **C** Solution 2D ¹⁹F{¹¹⁹Sn} J-HSQC NMR spectrum of the model toothpaste acquired with the pulse sequence shown in (**B**). Splittings due to ¹⁹F-¹¹⁹Sn one-bond J-couplings (¹J) are indicated.

plausibly be ascribed to $SnF_4$. However, we tentatively assign the −785 ppm ¹¹⁹Sn NMR signal to $SnF_4$ based on the much larger signal intensity observed in the 1D ¹¹⁹Sn spin echo NMR spectrum and the ¹⁹F chemical shifts observed in the 1D ¹⁹F spin echo NMR spectrum (Fig. 1B and S1). We suspect that the other unassigned signals are derived from impurities, such as hydrated phases or other polymorphic forms of $SnF_4$.

2D ¹⁹F{¹¹⁹Sn} J-HMQC NMR spectra of $SnF_2$ from supplier **b** reveals the expected correlations between all ¹⁹F and ¹¹⁹Sn NMR signals associated with $SnF_2$ (Fig. 1C, lower and Supplementary Fig. 3). The 2D ¹⁹F{¹¹⁹Sn} J-HMQC NMR spectrum of $SnF_2$ from supplier **a** reveals additional correlations between different ¹⁹F and ¹¹⁹Sn NMR signals that

were not observed for $SnF_2$ from supplier **b** (Fig. 1C, upper). Notably, two unique sets of ¹⁹F NMR signals of $SnF_2$ from supplier **a** resonate at significantly lower $\delta_{iso}$(¹⁹F) than expected for $SnF_2$; the $\delta_{iso}$(¹⁹F) is near that observed for $SnF_4$ (Supplementary Fig. 2). The lowest frequency ¹⁹F NMR signal correlates with a ¹¹⁹Sn that exhibits no observable CSA with a 25 kHz MAS frequency. Based on the low frequency ¹⁹F NMR signal and small ¹¹⁹Sn CSA, we assign these NMR signals to Sn(IV) fluoride impurities. We note that the observed ¹⁹F and ¹¹⁹Sn NMR signals are different than those assigned to $SnF_4$.

In summary, $SnF_2$ exhibits a significantly larger ¹¹⁹Sn CSA than $SnF_4$, illustrating that the ¹¹⁹Sn CSA is a good probe of the Sn oxidation

state, consistent with prior literature. Additionally, the [19]F chemical shifts for $SnF_2$ are much more positive than those for $SnF_4$, suggesting that [19]F NMR can also provide insight into the Sn oxidation state. Interestingly, impurities in the $SnF_2$ and $SnF_4$ samples from supplier **a** were detected in 1D [19]F and [119]Sn NMR spectra. Notably, a 2D [19]F{[119]Sn} *J*-HMQC spectrum of $SnF_2$ from supplier **a** revealed an Sn(IV) fluoride impurity. The observation of an Sn(IV)-based impurity is important because $SnF_2$-based toothpastes rely on maximum Sn(II) availability for optimal performance. Impure $SnF_2$ starting materials will lead to a less effective toothpaste. [19]F and [119]Sn MAS solid-state NMR spectroscopies are good tools to determine the purity of tin fluoride materials, which may be used as precursors in toothpaste products.

Solution [19]F and [119]Sn NMR Spectroscopy. We performed room temperature solution [19]F and [119]Sn NMR spectroscopy experiments on a model toothpaste to initially probe all Sn and F atoms before studying commercially available toothpastes with DNP-enhanced [119]Sn NMR spectroscopy. The model toothpaste consisted of *ca.* 2 wt% $SnF_2$ in a 1:2 mixture of $D_2O$:glycerol$_{d-8}$; the solution was prepared *ca.* 1 month before running NMR experiments. The use of $SnF_2$ in $D_2O$:glycerol$_{d-8}$ makes this a simplified version of most $SnF_2$-based toothpastes, allowing for an easier assessment of the Sn speciation before studying more complex commercial toothpaste formulations. We note that the solution [19]F and [119]Sn NMR experiments suggest that the majority of F atoms are dissociated from Sn. As discussed in more detail below, we assume that only Sn atoms fully dissociated from F will be primarily observed in the DNP-enhanced [119]Sn solid-state NMR spectra because we lack the capability to decouple [19]F. The sizeable [19]F heteronuclear couplings for F coordinated tin ions could lead to reduced [119]Sn homogeneous transverse relaxation time constants ($T_2'$) and low [119]Sn CPMG NMR signal intensities.

A 1D [19]F solution NMR spectrum of the model toothpaste reveals primarily two [19]F NMR signals at −88.3 ppm and −157.4 ppm (Fig. 2A). The broad hump from −150 ppm to −200 ppm is a probe background [19]F NMR signal (Supplementary Fig. 4). Closer examination of the −157.4 ppm [19]F NMR signal reveals three set of doublets centered around the isotropic NMR signal which correspond to 1-bond *J*-couplings of [19]F-[119]Sn (1571 Hz), [19]F-[117]Sn (1505 Hz) and [19]F-[115]Sn (1385 Hz) ([1]*J*; Fig. 2C, upper and Supplementary Fig. 4). The integral of each doublet matches with the corresponding Sn isotopic abundances and the *J*-couplings scale with the gyromagnetic ratios of the tin isotopes. We recorded a 2D [19]F{[119]Sn} *J*-HMQC solution NMR spectrum to probe all F atoms bonded to Sn (Supplementary Fig. 5). Only the [19]F NMR signals near −155 ppm were observed in the 2D *J*-HMQC NMR spectrum. Therefore, we assign the [19]F NMR signal at −90 ppm to free fluoride ions. Integration of the [19]F NMR signals reveals that at least *ca.* 92% of F is present as ions within the $D_2O$/glycerol$_{d-8}$ mixture (Fig. 2A). We note that the population of solvated F ions is likely even higher as the [19]F transmitter was on resonance with the **F**-Sn [19]F NMR signal and due to overlap of the −157.4 ppm signal with the probe background [19]F NMR signals (see Methods).

The [19]F NMR signal assigned to an **F**-Sn species resonates at a similar shift as was observed for $SnF_4$ (Fig. 2A). To confirm the origin of the −157.4 ppm [19]F NMR signal we recorded a 2D [19]F{[119]Sn} *J*-HSQC solution NMR spectrum using the modified pulse sequence shown in Fig. 2B. The modified pulse sequence does not have a central refocusing (π) pulse applied to the [19]F spins during [119]Sn $t_1$ evolution, which causes evolution of [19]F-[119]Sn *J*-couplings and [119]Sn chemical shifts in the indirect dimension, resulting in a multiplet dependent on the number of attached F atoms to Sn. The modified 2D [19]F{[119]Sn} *J*-HSQC solution NMR spectrum reveals a septet like pattern in the [119]Sn dimension and a doublet in the [19]F dimension ([119]Sn decoupling was not performed during acquisition, Fig. 2C). Note the central line of the septet pattern is absent because the central line of the multiplet arises from [119]Sn spins coupled to 3 [19]F spin-up and 3 [19]F spin-down leading to an effective *J*-coupling of 0 Hz. The observed [119]Sn septet matches exactly

to that of a numerical simulation for Sn attached to six F atoms; the three doublets exhibit splittings at 2, 4, or 6 times [1]$J_{Sn-F}$ and the intensities are consistent with an $AX_6$ spin system determined from a modified Pascal triangle[82]. Therefore, the [19]F NMR signal at −157.4 ppm is assigned to the $(SnF_6)^{-2}$ anions.

In summary, integration of the 1D [19]F solution NMR spectrum suggests at most *ca.* 8% of F is directly bonded to Sn (Fig. 2A). However, assuming that the two [19]F NMR signals correspond to free fluoride anions (F−) and $(SnF_6)^{-2}$, then only *ca.* 2% of Sn within the model toothpaste contains F bonds, consistent with a 1D [119]Sn solution NMR spectrum that does not show appreciable amounts of **Sn**-F species (Supplementary Fig. 6). This observation is important because it implies essentially all Sn atoms are observable in the DNP [119]Sn solid-state NMR experiments. Sn species exhibiting F bonds would exhibit significant [119]Sn NMR signal attenuation in DNP solid-state NMR due to large [19]F-[119]Sn dipolar couplings that will not be effectively averaged in the absence of [19]F heteronuclear decoupling.

Dynamic Nuclear Polarization [119]Sn Solid-State NMR Spectroscopy. We performed dynamic nuclear polarization (DNP) enhanced [119]Sn solid-state NMR spectroscopy on the model toothpaste, a commercially available preventative gel (**pg1**) and four commercially available toothpastes (**t1-t4**). The weight loadings of $SnF_2$ in the model toothpaste, **pg1** and **t1-t4** are *ca.* 2 wt%, 0.40 wt% or 0.454 wt%, corresponding to absolute Sn loadings of *ca.* 1.5 wt%, 0.30 wt% or 0.34 wt%, respectively. **pg1** contains primarily glycerol, similar to the model toothpaste (glycerol and water, Supplementary Table 2). **t1** and **t3** are glycerol-based, while **t2** and **t4** are water and glycerol and/or sorbitol-based. In addition the toothpastes contain other ingredients, such as abrasives (Table S2). **t4** included $SnCl_2$ as an $SnF_2$ stabilizer. Samples were prepared for DNP experiments by directly dissolving the AMUPol biradical at a final concentration of 11 mM within the toothpastes (see *Methods*). For the model toothpaste, $H_2O$ was added to increase DNP enhancements (10:30:60 ratio of $H_2O$:$D_2O$:gylcerol$_{d-8}$). All DNP experiments were performed immediately after biradical addition. Prolonged storage of the DNP samples in a −20 °C freezer for days resulted in no DNP enhancements, likely due to reduction of the nitroxide biradical caused by conversion of Sn(II) to Sn(IV). In the absence of dissolved $SnF_2$, the nitroxide biradical water glycerol solutions retain their DNP enhancements after months of storage in a freezer, consistent with reaction of Sn(II) with the nitroxide radicals causing the loss in DNP enhancements.

[1]H → [13]C cross-polarization magic-angle spinning (CPMAS) DNP enhancements (ε) of the model toothpaste and **pg1** were *ca.* 84 and 105, respectively (Fig. 3A and Supplementary Fig. 7). This means that acquisition of solid-state NMR spectra with the same signal-to-noise (SNR) ratio would take more than a thousand times longer to acquire without MW irradiation (i.e., no DNP). The DNP enhancement (ε) measured with [1]H → [119]Sn CP-CPMG experiments was estimated to be between 12–45 for the model toothpaste and **t1-t4** (Fig. 3B and Supplementary Fig. 8). However, a [119]Sn NMR spectrum could not be recorded without MW irradiation in a reasonable amount of time (*ca.* 1 h). Therefore, we are likely under-estimating the [119]Sn DNP enhancements. The [119]Sn DNP enhancements are likely similar to those measured for [13]C since in both [119]Sn and [13]C cross-polarization NMR experiments, the magnetization is derived from the same bath of [1]H spins. The high DNP enhancements enables the acquisition of 2D [1]H-[119]Sn CP heteronuclear correlation (HETCOR) and magic-angle turning (MAT) NMR spectra of all samples.

A 2D [1]H → [13]C CP-HETCOR NMR spectrum of the model toothpaste reveals correlations between the -O**H** [1]H NMR signals of glycerol or $H_2O$ and the [13]C NMR signals of glycerol (Fig. 3C). The glycerol C**H**$_x$ [1]H NMR signals were not observed as the glycerol was fully deuterated. A 2D [1]H → [13]C CP-HETCOR NMR spectrum of preventative gel **1** reveals the expected correlations between all the [1]H and [13]C NMR signals of the C**H**$_x$ and -OH groups of the glycerol solvent (Supplementary Fig. 9A).

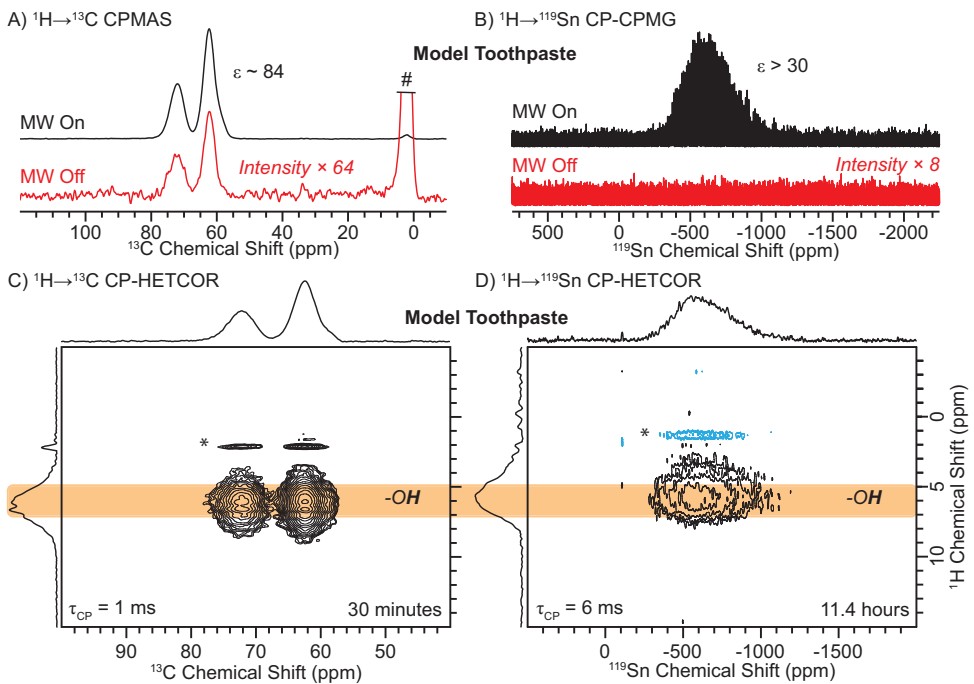

**Fig. 3 | DNP-enhanced ¹³C and ¹¹⁹Sn solid-state NMR spectra.** Comparison of (**A**) ¹H → ¹³C CPMAS and (**B**) ¹H → ¹¹⁹Sn CP-CPMG NMR spectra of the model toothpaste recorded (black) with or (red) without microwave (MW) irradiation. The DNP enhancements (ε) are given in the figure. # denotes the truncated glycerol signal. DNP-enhanced (**C**) ¹H → ¹³C and (**D**) ¹H → ¹¹⁹Sn 2D CP-HETCOR NMR spectra of the model toothpaste. Spectra were recorded with a 10 kHz MAS frequency, eDUMBO$_{1-22}$ ¹H homonuclear decoupling during ¹H indirect dimension evolution time, and CPMG detection of ¹¹⁹Sn. The orange band illustrates that the same ¹H chemical shift is present in both ¹H NMR spectra. CP contact times (τ$_{CP}$) and total experiment times are indicated. Asterisks denote quadrature artifacts that occur at ¹H transmitter frequency.

Interestingly, 2D ¹H → ¹¹⁹Sn CP-CPMG HETCOR NMR spectra of the model toothpaste and **pg1** both display correlations between all ¹H NMR signals observed in the 2D ¹H → ¹³C CP-HETCOR NMR spectra with a broad ¹¹⁹Sn NMR signal (Fig. 3D and Supplementary Fig. 9B). The observed ¹H-¹¹⁹Sn correlations reveal that the Sn is present as ions that are likely solvated by H$_2$O and/or glycerol. The interaction between Sn(II) and glycerol was also observed in liquid chromatography mass spectrometry (LCMS) experiments (Supplementary Fig. 10).

1D ¹H → ¹¹⁹Sn CP-CPMG NMR spectra of all samples were acquired with multiple ¹¹⁹Sn transmitter offsets (VOCS acquisition[83,84]) due to the large breadth of the ¹¹⁹Sn NMR signals and relatively low ¹¹⁹Sn NMR excitation bandwidth (Supplementary Fig. 11). The 1D ¹H → ¹¹⁹Sn CP-CPMG NMR spectra of the model toothpaste and **pg1** reveal primarily a broad ¹¹⁹Sn NMR signal *ca.* 1500 ppm in breadth, a larger range than was observed for SnF$_2$ (*ca.* 1000 ppm). Likewise, the 1D ¹H → ¹¹⁹Sn CP-CPMG NMR spectra of **t1-t4** reveal broad ¹¹⁹Sn NMR signals that cover a range of *ca.* 1000–1500 ppm in addition to sharper ¹¹⁹Sn NMR features. The broad ¹¹⁹Sn NMR signals likely correspond to Sn(II) species whereas the sharper features likely correspond to Sn(IV). However, identification of Sn(II) and Sn(IV) species is fairly ambiguous from the 1D NMR spectra alone. The 1D ¹¹⁹Sn MAS solid-state NMR spectra are likely broad and featureless because of the presence of isotropic chemical shift distributions, which are typical of disordered and amorphous systems. There is probably a distribution in the number of water, hydroxide, and glycerol (or glyceroxide) molecules coordinated to each tin ion.

To resolve Sn(II) from Sn(IV) species based on their CSA, we recorded DNP-enhanced 2D ¹¹⁹Sn adiabatic magic angle turning (aMAT) NMR spectra of all samples (Fig. 4 and S12–S15)[85-88]. These 2D NMR experiments required only *ca.* 6 h for the model toothpaste (Sn loading ~ 1.5 wt%) and *ca.* 16–17 h for **pg1** and **t1-t4** (Sn loading = 0.3 or 0.34 wt%, respectively). In a MAT NMR experiment, an isotropic NMR spectrum free of spinning sidebands (indirect dimension) is correlated

with its corresponding anisotropic MAS NMR spectrum (direct dimension). Therefore, the anisotropic NMR spectra extracted at specific isotropic chemical shifts (δ$_{iso}$) can be easily fit to determine the span (Ω) and skew (κ) because the δ$_{iso}$ is known.

2D ¹¹⁹Sn aMAT NMR spectra of the model toothpaste and **pg1** are near identical and reveal broad isotropic ¹¹⁹Sn NMR spectra in the region of *ca.* –600 to –1000 ppm (Fig. 4A and S12A). The broad isotropic ¹¹⁹Sn NMR spectra illustrate that there are large distributions in the ¹¹⁹Sn δ$_{iso}$, likely due to differences in the Sn ion coordination from the solvent matrix. We note that the center of the isotropic ¹¹⁹Sn NMR spectrum of the model toothpaste appears at the same ¹¹⁹Sn chemical shift observed in solution, however, the breadth of the signal is much narrower at room temperature due to dynamics of the Sn ions in solution (Supplementary Fig. 6). Anisotropic ¹¹⁹Sn NMR spectra extracted from the isotropic ¹¹⁹Sn dimension reveals primarily Sn sites with a Ω of *ca.* 1200 ppm and a κ of +0.7, which are assigned to Sn(II) species based on the large CSA (Fig. 4B and S12B). However, anisotropic ¹¹⁹Sn NMR spectra extracted at higher ¹¹⁹Sn δ$_{iso}$ of *ca.* –750 ppm or –800 ppm to –600 ppm for the model toothpaste or **pg1**, respectively, show an increased intensity for the isotropic (center band) NMR signals. An increase in intensity for the isotropic NMR signal reveals there are additional sites with spans of 150 ppm or less, but with identical ¹¹⁹Sn isotropic chemical shift to sites that have a larger span; the small CSA sites should correspond to Sn(IV). Therefore, the 2D ¹¹⁹Sn aMAT NMR spectra clearly reveal both Sn(II) and Sn(IV) sites based on their CSA. We note that the experimental anisotropic NMR spectra show distorted signal intensities at lower ¹¹⁹Sn chemical shifts due to limited ¹¹⁹Sn NMR excitation bandwidth. Nevertheless, Ω and κ can still be accurately determined by fitting the most intense spinning sidebands with the isotropic shift as a fixed constraint.

DNP-enhanced 2D ¹¹⁹Sn aMAT NMR spectra of **t1, t2,** and **t3** are near identical and display three relatively sharp isotropic ¹¹⁹Sn NMR

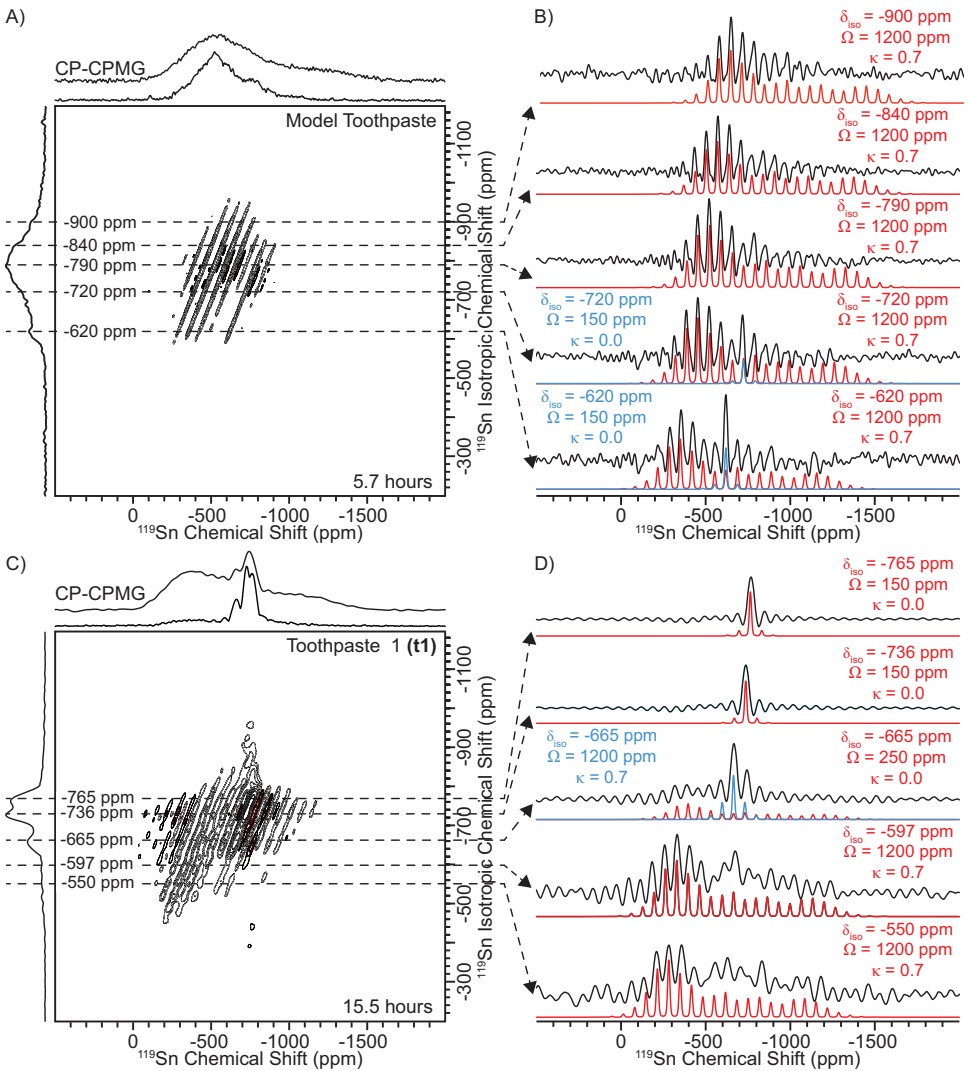

**Fig. 4 | DNP-enhanced $^{119}$Sn aMAT NMR spectra.** 2D $^{119}$Sn aMAT NMR spectra of (**A**) the model toothpaste and (**C**) **t1** acquired with a 10 kHz MAS frequency, $^1$H → $^{119}$Sn CP at the start of the experiment, and CPMG for $^{119}$Sn detection. **B**, **D** $^{119}$Sn solid-state NMR spectra extracted from the 2D aMAT NMR spectra at the indicated $^{119}$Sn isotropic chemical shifts ($\delta_{iso}$). Analytically simulated spectra are shown (colored) below the (black) experimental MAS spectra. The values of the isotropic chemical shift ($\delta_{iso}$), span ($\Omega$) and skew ($\kappa$) used in the analytical simulations are indicated next to each row.

signals at *ca.* −762 to −775 ppm, −732 to −744 ppm and −665 ppm, in addition to a broad isotropic $^{119}$Sn NMR signal from *ca.* −500 to −600 ppm (Fig. 4C, S13A and S14A). The relatively sharp $^{119}$Sn NMR signals at *ca.* −762 to −775 ppm and −732 to −744 ppm clearly show intense isotropic $^{119}$Sn NMR signals, which are assigned to Sn(IV) species based on the small $\Omega$ (*ca.* 150 ppm; Fig. 4A, S13B and S14B). There are additional weak sidebands associated with Sn that have spans on the order of *ca.* 1200 ppm, suggesting some Sn(II) species are present at these isotropic shifts. Anisotropic $^{119}$Sn NMR spectra extracted at more positive $^{119}$Sn $\delta_{iso}$ of *ca.* −665 ppm and −600 ppm reveal significantly more intense broad $^{119}$Sn NMR spectra with spans of *ca.* 1200 ppm (Fig. 4A, S13B and S14B). At the lower $^{119}$Sn $\delta_{iso}$ of *ca.* −665 ppm, the isotropic NMR signal has increased signal intensity, consistent with additional small CSA sites ($\Omega \approx$ 150−250 ppm). However, the more positively shifted isotropic $^{119}$Sn NMR signals are clearly primarily associated with Sn(II) sites. We note that the isotropic $^{119}$Sn NMR spectra are not representative of the Sn(II) and Sn(IV) populations due to differences in MAT efficiencies for high or low CSA sites, respectively. The similarities in the 2D $^{119}$Sn aMAT spectra of **t1** and **t3** are not surprising because they are both primarily glycerol-based. However, the similarities between the MAT NMR spectrum of **t2** with the MAT

NMR spectra of **t1** and **t3** are interesting because **t2** contains a significant amount of water in addition to glycerol.

Interestingly, the 2D $^{119}$Sn aMAT NMR spectrum of **t4** is significantly different from that of **t1-t3** (Supplementary Fig. 15A). As mentioned above, **t4** contains primarily water and sorbitol (similar to **t2**), in addition to SnCl$_2$ as a SnF$_2$ stabilizer (Table S2). The isotropic $^{119}$Sn NMR spectrum of **t4** shows primarily three isotropic $^{119}$Sn NMR signals at *ca.* −665 ppm, −550 ppm and −475 ppm (Supplementary Fig. 15B). Similar $^{119}$Sn isotropic NMR signals were observed for **t1-t3**, however, for **t4** the $^{119}$Sn isotropic NMR signals at *ca.* −665 and −550 ppm correspond to predominantly small CSA sites ($\Omega \approx$ 150−220 ppm; Supplementary Fig. 15B). The $^{119}$Sn isotropic NMR signal at *ca.* −475 ppm clearly shows sites with both large and small CSA ($\Omega \approx$ 220 ppm and 1100 ppm).

With knowledge of all $^{119}$Sn chemical shift tensors, the 1D $^1$H → $^{119}$Sn CP-CPMG NMR spectra of all samples could be analytically simulated (Fig. 5 and Supplementary Table 3). The 2D $^{119}$Sn aMAT NMR spectra revealed large distributions in the $^{119}$Sn $\delta_{iso}$. Therefore, we fit the 1D $^1$H → $^{119}$Sn CP-CPMG NMR spectra to sites containing large amounts of Gaussian line broadening to represent the distributions in the isotropic chemical shifts.

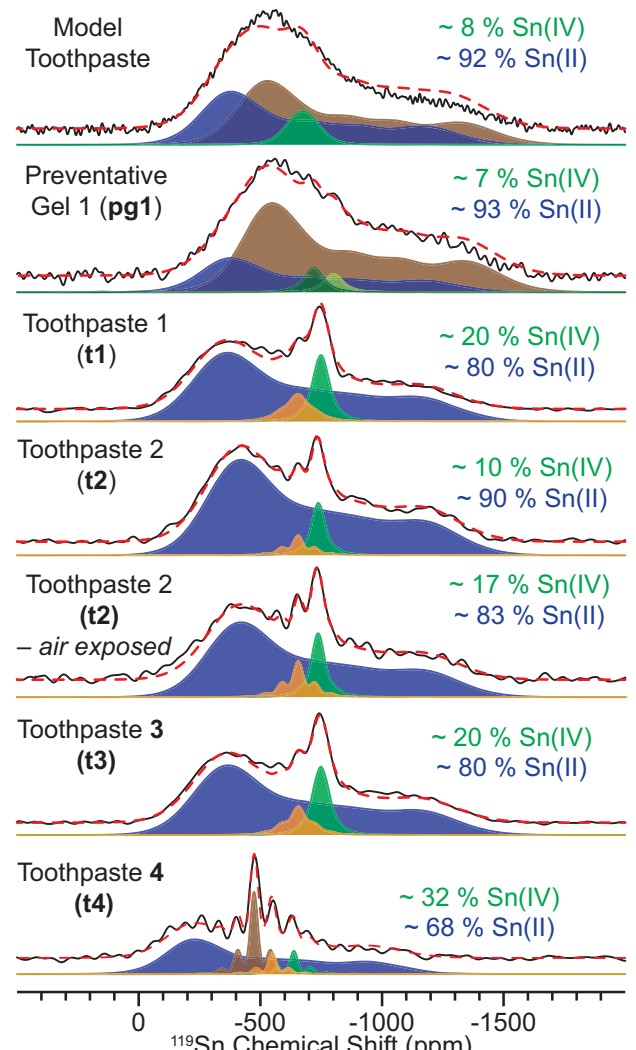

**Fig. 5 | DNP-enhanced MAS $^1H \rightarrow {}^{119}Sn$ CP-CPMG solid-state NMR spectra.** (upper to lower) $^1H \rightarrow {}^{119}Sn$ CP-CPMG NMR spectra for the model toothpaste, **pg1**, and **t1-t4**. Multiple CP-CPMG NMR spectra were recorded with different $^{119}Sn$ transmitter frequencies (i.e., VOCS style acquisition, Supplementary Fig. 11). All spectra were recorded with a 10 kHz MAS frequency and a sample temperature of approximately 110 K. Total fits are indicated by red dashed lines. Solid blue, brown, orange, and green peaks correspond to the individual sites used in fitting.

The model toothpaste and **pg1** display similar 1D $^1H \rightarrow {}^{119}Sn$ CP-CPMG NMR spectra, where the populations of Sn(II) were determined to be *ca*. 92 or 93%, respectively (Fig. 5). **t1-t3** also display similar 1D $^1H \rightarrow {}^{119}Sn$ CP-CPMG NMR spectra (Fig. 5). The populations of Sn(II) were determined to be *ca*. 80% for **t1** and **t3** and 90% for **t2**. We also recorded a 1D $^1H \rightarrow {}^{119}Sn$ CP-CPMG NMR spectrum of **t2** after allowing the toothpaste to dry out and exposing it to air over the course of 1 day (Fig. 5). Interestingly, the population of Sn(II) decreased from *ca*. 90 to 83%, resulting from oxidation of Sn(II) to Sn(IV) from $O_2$ in the atmosphere. This experiment was an important control that confirms our hypothesis that the $^{119}Sn$ NMR signals of Sn(IV) primarily have small spans, while those associated with Sn(II) have larger spans. Consistent with the differences observed in the 2D $^{119}Sn$ aMAT NMR spectrum, **t4** displays a different 1D $^1H \rightarrow {}^{119}Sn$ CP-CPMG NMR spectrum with a significantly lower amount of Sn(II) (*ca*. 68%; Fig. 5). We note that the relative populations of Sn(II) for **t1-t4** determined here are generally consistent with prior measurements made in Sn K-edge X-ray absorption studies[17,34].

There are three main mechanisms that can lead to inaccurate Sn(II) and Sn(IV) populations determined from the 1D $^1H \rightarrow {}^{119}Sn$ CP-

CPMG NMR spectra: (1) differences in DNP enhancement, (2) difference in $^1H \rightarrow {}^{119}Sn$ CP dynamics, and (3) differences in $^{119}Sn$ refocused transverse relaxation time constants ($T_2'$). DNP enhancements for Sn(II) and Sn(IV) should be identical since 2D $^1H \rightarrow {}^{119}Sn$ CP-HETCOR NMR spectra revealed that Sn is present as ions within the solvent matrix and $^1H$ spin diffusion should distribute the DNP enhanced $^1H$ polarization homogeneously across the frozen solution (Fig. 3D and Supplementary Fig. 9B). $^1H \rightarrow {}^{119}Sn$ CP dynamics are likely similar for Sn(II) and Sn(IV) sites since they are both likely coordinated by water, hydroxide ions and/or glycerol molecules. However, $^1H \rightarrow {}^{119}Sn$ CP is likely less efficient for sites with high CSA because the CSA is comparable to or larger than the RF field used for the $^{119}Sn$ spin-lock pulse. From this perspective, the Sn(II) populations are likely a lower bound. To assess differences in $^{119}Sn$ $T_2'$, we investigated the effect that the number of CPMG echoes used during that acquisition of $^1H \rightarrow {}^{119}Sn$ CP-CPMG NMR spectra of the model toothpaste and **t1** had on the determined Sn(II) and Sn(IV) populations (Supplementary Fig. 16). $^1H \rightarrow {}^{119}Sn$ CP-CPMG NMR spectra processed with 1 to 100 spin echoes in the CPMG train reveal near identical populations of Sn(II) and Sn(IV), confirming that the $^{119}Sn$ $T_2'$ must be similar for all species. The observation of similar $^{119}Sn$ $T_2'$ for all Sn species is also consistent with minimal Sn sites exhibiting F bonds, as those sites would exhibit a shorter $^{119}Sn$ $T_2'$. Therefore, analytical simulations of the $^1H \rightarrow {}^{119}Sn$ CP-CPMG NMR spectra likely reveal relatively accurate populations of Sn(II) and Sn(IV), where the population of Sn(II) should be taken as a lower bound due to differences in CP efficiencies.

In conclusion, we applied dynamic nuclear polarization (DNP) enhanced $^{119}Sn$ solid-state NMR spectroscopy to determine the Sn oxidation state and speciation within commercially available $SnF_2$-based toothpastes that contain loadings of less than 0.5 wt%. We first obtained room-temperature $^{19}F$ and $^{119}Sn$ solid-state NMR spectra of $SnF_2$ and $SnF_4$. These experiments confirmed Sn(II) exhibits a near order of magnitude larger span than that of Sn(IV), consistent with prior literature. NMR studies of $SnF_2$ purchased from two different suppliers revealed a significant amount of Sn and F-based impurities in one of the samples. Notably, 2D $^{19}F\{^{119}Sn\}$ *J*-HMQC NMR spectra revealed that the impure $SnF_2$ sample contains Sn(IV) fluoride-based impurities. $^{19}F$ and $^{119}Sn$ solid-state NMR are good probes of the purity of $SnF_2$ precursors used in the production of toothpastes. Solution $^{19}F$ and $^{119}Sn$ NMR studies on a model toothpaste consisting of *ca*. 2 wt% $SnF_2$ in a 50:50 mixture of $D_2O$:glycerol$_{d-8}$ suggested that only *ca*. 2% of the dissolved Sn ions contain F bonds. DNP experiments of model and commercially available toothpastes were enabled by directly mixing the DNP polarizing agent (AMUPol biradical) within the toothpaste. Importantly, the sensitivity gains offered by DNP enabled detection of $^{119}Sn$ NMR signals from toothpastes with loadings of 0.34 wt% Sn. 2D $^1H \rightarrow {}^{13}C$ and $^1H \rightarrow {}^{119}Sn$ CP-HETCOR NMR spectra of the model toothpaste and **pg1** suggested that all Sn is present as ions that are solvated by water, hydroxide anions and/or glycerol. 1D $^1H \rightarrow {}^{119}Sn$ CP-CPMG NMR spectra of all toothpastes revealed broad $^{119}Sn$ NMR spectra, with some additional sharper features. Acquisition of 2D $^{119}Sn$ magic-angle turning (MAT) NMR spectra of all samples allowed for the unambiguous identification of Sn(II) and Sn(IV) species based on their CSA. With knowledge of the $^{119}Sn$ chemical shift tensor parameters, 1D $^1H \rightarrow {}^{119}Sn$ CP-CPMG NMR spectra were fit to estimate the populations of Sn(II) and Sn(IV) within the toothpastes. Notably, three of the four commercially available toothpastes contained at least 80% Sn(II), whereas one of the toothpaste contained a significantly higher amount of Sn(IV).

We have demonstrated that DNP-enhanced $^{119}Sn$ solid-state NMR spectroscopy is an ideal technique to probe the Sn speciation with commercially available toothpastes. The determination of the Sn(II) and Sn(IV) populations within commercially available toothpastes is important both to assess the quality of current formulations and to develop new and improved formulations. Increasing the amount of

Sn(II) should increase the antimicrobial properties of $SnF_2$-based toothpastes. We observed that both glycerol-based toothpastes (**t1** and **t3**) and toothpastes containing high amounts of water and glycerol (**t2**) can exhibit high amounts of Sn(II) (*ca.* 80–90%). However, the Sn(II) is readily oxidized to Sn(IV) after prolonged air-exposure. More detailed studies on the specific coordination of Sn and their interactions with other common toothpaste ingredients are on-going in our labs. By better understanding how common toothpaste ingredients interact with Sn, and specifically Sn(II), DNP-enhanced $^{119}Sn$ solid-state NMR spectroscopy will enable the rational design and development of next-generation $SnF_2$-based toothpastes that exhibit increased Sn(II) availability and long-term oxidation stability.

## Methods

Samples of $SnF_4$ and $SnF_2$ (supplier **a**) were purchased from Sigma Aldrich, Inc. $SnF_2$ supplier **b** corresponds to the $SnF_2$ that is used in commercial toothpaste products. AMUPol was purchased from CortecNet Inc. Glycerol-$d_8$ and $D_2O$ were purchased from Sigma-Aldrich Inc. All materials were used as received without further purification. Two different model toothpastes were prepared for solution NMR experiments and DNP solid-state NMR experiments, respectively. A micropipette was used to transfer the water-glycerol solutions. However, due to the high viscosity of the glycerol solutions, an analytical balance was used to weigh the amount of solution transferred to ensure that the desired volumes of water, glycerol and water-glycerol solutions were transferred. For the solution NMR experiments, a 2 wt% $SnF_2$ solution was prepared by dissolving 7.1 mg of $SnF_2$ in 347.6 mg of a 1:2 volume fraction solution of $D_2O$:glycerol-$d_8$. The solution was stored for 1 month at room temperature before running solution NMR experiments. The model toothpaste for DNP solid-state NMR experiments was prepared by first making approximately 200 μL (260 mg) of a stock solution of 10:30:60 volume fraction $H_2O$:$D_2O$:glycerol-$d_8$. 2.2 mg of $SnF_2$ was weighed out and 107.9 mg of the stock $H_2O$:$D_2O$:glycerol-$d_8$ solution was added to give a final $SnF_2$ concentration of 2 wt%. The $SnF_2$ was allowed to dissolve over the course of approximately 2 h. This solution as then stored in a freezer until it was needed. To prepare samples for DNP experiments, a few mg of the AMUPol biradical were then weighed out in a glass vial and $SnF_2$ $H_2O$:$D_2O$:glycerol-$d_8$ stock solution was then added to obtain a final AMUPol concentration of 10 ± 2 mM. Approximately 20 μL of this solution was then transferred to a sapphire DNP rotor which was then capped with a silicone plug and a zirconia drive cap. The packed rotor was then transferred into the pre-cooled DNP probe within 20 min to limit oxidation of the $Sn^{2+}$ ions by reaction with AMUPol. AMUPol was used for DNP experiments because this radical gives high DNP enhancements and sensitivity gains at 9.4 T for water-glycerol based mixtures[66].

Samples of commercial toothpastes were prepared for DNP experiments by directly mixing the AMUPol biradical within the toothpastes to obtain a final AMUPol concentration of *ca.* 10 mM. A typical sample preparation of the commercial toothpastes for DNP consisted of weighing out the AMUPol biradical in a vial (*ca.* 1.3–2.3 mg), adding the proper amount of toothpaste to reach a concentration of *ca.* 10 mM, and then vigorously stirring the toothpaste for *ca.* 10 min to ensure the radical was homogenously mixed throughout the toothpaste. The densities of the toothpastes were assumed to be *ca.* 1.3 g cm$^{-3}$. We note that samples of toothpastes **1**–**4** were taken from the middle of a fresh toothpaste tube, while preventative gel **1** was taken from the top of a fresh tube. During the mixing step, the vial was periodically held under a stream of warm water for short time periods (a maximum time of *ca.* 10 s) to decrease the viscosity of the toothpaste and facilitate better mixing and dissolution of the radical. Once the radical was homogenously mixed with the toothpaste, the sample was immediately packed into a 3.2 mm sapphire rotor. The sapphire rotor was sealed with a silicone soft plug and capped with a zirconia drive cap. All DNP samples were prepared immediately before performing NMR experiments. The maximum time it took to prepare the sample and insert the rotor into the spectrometer was *ca.* 20–30 min. We note that prolonged storage (1–2 weeks) of the prepared samples at *ca.* 0 °C gave no DNP enhancements due to reduction of the biradical, presumably caused by oxidation of Sn(II) to Sn(IV).

Room temperature solution $^{19}F$ and $^{119}Sn$ solution NMR experiments were performed on the model toothpaste and recorded on a 9.4 T ($v_0(^1H) = 400$ MHz) Bruker standard-bore magnet equipped with a AVANCE NEO console and a liquid-$N_2$ cooled Bruker Prodigy HXY NMR probe. $^{19}F$ and $^{119}Sn$ chemical shifts were referenced to $CCl_3F$ and $SnMe_4$, respectively, with $D_2O$ as the lock signal. The $^{19}F$ π/2 and π pulses were 15 and 30 μs in duration, corresponding to a 16.7 kHz radio frequency (RF) field. We note that the $^{19}F$ NMR spectrum was recorded with the $^{19}F$ transmitter on resonance with the $SnF_6^{-2}$ $^{19}F$ NMR signal. The $^{19}F$ NMR signals of the free F ions were *ca.* 26 kHz away from the $^{19}F$ transmitter. The $^{19}F$ spin echo NMR spectrum was recorded with 100 μs delays on each side of the $^{19}F$ π pulse. The $^{19}F$ solution NMR spectra were acquired with different recycle delays to ensure that quantitative relative peak intensities were obtained. Recycle delays used for solution NMR experiments are given in Supplementary Table 4. The $^{119}Sn$ π/2 and π pulses were 12.5 μs and 25 μs in duration, corresponding to a 20 kHz RF field. The $^{119}Sn$ spin echo NMR spectrum was recorded with 10 μs echo delays on each side of the $^{119}Sn$ π pulse.

Room Temperature solid-state NMR spectroscopy experiments were performed with a 9.4 T ($v_0(^1H) = 400$ MHz) Bruker wide-bore magnet equipped with a Bruker Avance III HD console and a 2.5 mm HXY magic-angle spinning (MAS) NMR probe configured in triple resonance mode. We note that the $^{19}F$ and $^{119}Sn$ match was relatively poor (*ca.* 30 and 60% for $^{19}F$ and $^{119}Sn$, respectively) when tuned simultaneously to $^{19}F$ and $^{119}Sn$ on the $^1H$ and X channel, respectively. The magnetic field was referenced to 1% tetramethyl silane (TMS) in $CDCl_3$ with adamantane ($\delta(^1H) = 1.71$ ppm) as a secondary chemical shift reference. $^{19}F$ and $^{119}Sn$ chemical shifts were indirectly referenced to $CCl_3F$ and $SnMe_4$, respectively, using the previously published IUPAC recommended relative NMR frequencies[89].

The $^{19}F$ π/2 and π pulses were 4 and 8 μs in duration, corresponding to a 62.5 kHz radio frequency (RF) field. The $^{119}Sn$ π/2 and π pulses were 3.5 and 7 μs in duration, corresponding to a 71 kHz RF field. 2D $^{19}F\{^{119}Sn\}$ *J*-based heteronuclear multiple quantum correlation (*J*-HMQC) experiments were recorded with the arbitrary indirect dwell (AID) HMQC pulse sequence[88]. SPINAL-64 heteronuclear decoupling with a 50 kHz $^{19}F$ RF field was performed during the acquisition of $^{119}Sn$ NMR signals[90]. $^{119}Sn$ and $^{19}F$ solid-state NMR spectra were acquired at multiple MAS frequencies to confirm the assignment of isotropic and sideband NMR signals (Supplementary Fig. 17).

DNP-enhanced solid-state NMR spectroscopy experiments were performed with a 9.4 T ($v_0(^1H) = 400$ MHz) Bruker wide-bore magnet equipped with a 263 GHz gyrotron, a Bruker AVANCE III console and a 3.2 mm HXY MAS DNP NMR probe[67]. MAS solid-state NMR experiments were performed with a sample temperature of ca. 110 K.

The magnetic field was referenced to 1% TMS in $CDCl_3$ with the $^1H$ shift of the silicone soft plug ($\delta(^1H) = 0.24$ ppm) as a secondary chemical shift reference. The $^1H$ shift of the silicone soft plug was determined based on the $^1H$ shift of frozen tetrachloroethane (TCE, $\delta(^1H) = 6.2$ ppm). $^{13}C$ and $^{119}Sn$ shifts were indirectly referenced to $SiMe_4$ or $SnMe_4$, respectively, using the previously published IUPAC recommended relative NMR frequencies[89]. All NMR spectra were initially processed and referenced with the Bruker Topspin 3.6.1 software. Carr-Purcell Meiboom-Gill (CPMG) echo trains were co-added using the NUTs NMR software (Acorn, Inc.). The $^{119}Sn$ solid-state NMR spectra were analytically fit using the open-source ssNake NMR software[91].

All experimental NMR parameters (MAS frequency, recycle delay ($\tau_{rec.\ delay}$), number of scans, $t_1$ dwell ($\Delta t_1$), $t_1$ TD points, $t_1$ acquisition

time ($t_1$ AQ), CP/$J$-evolution durations ($\tau_{CP/J\text{-evolv.}}$) and total experimental times are given in Supplementary Table 4. DNP NMR experiments were performed with the NMR probe configured in either HXY triple-resonance mode (tuned to $^1$H-$^{119}$Sn-$^{13}$C) or HX double-resonance mode (tuned to $^1$H-$^{119}$Sn). In all probe configurations, the $^1$H π/2 and π pulse lengths were 2.5 and 5 µs in duration, corresponding to a 100 kHz RF field. The $^{13}$C π/2 and π pulse lengths were 4 and 8 µs in duration, corresponding to a 62.5 kHz RF field. In triple-resonance HXY mode, the $^{119}$Sn π/2 and π pulse lengths were 4 and 8 µs in duration, corresponding to a 62.5 kHz RF field. In double-resonance HX mode, the $^{119}$Sn π/2 and π pulse lengths were 3 and 6 µs in duration, corresponding to an 83.3 kHz RF field. $^1$H → $^{13}$C cross-polarization (CP) was achieved with a 10 kHz MAS frequency with simultaneous $^1$H and $^{13}$C spin-lock pulses with RF fields of *ca.* 62 kHz (ramped from 56–62 kHz) and 64 kHz, respectively. In triple-resonance HXY mode (10 kHz MAS frequency), $^1$H → $^{119}$Sn CP was achieved with simultaneous $^1$H and $^{119}$Sn spin-lock pulses with RF fields of *ca.* 72 kHz (ramped from 65–72 kHz) and 56 kHz, respectively. In double-resonance HX mode (10 kHz MAS frequency), $^1$H → $^{119}$Sn CP was achieved with simultaneous $^1$H and $^{119}$Sn spin-lock pulses with RF fields of *ca.* 76 kHz (ramped from 69 – 76 kHz) and 80 kHz, respectively. Optimization of the $^1$H → $^{119}$Sn CP contact time showed that 6 ms was optimal.

All $^{119}$Sn NMR spectra were acquired with CPMG detection to increase sensitivity. π/2 (1D NMR spectra of **t1**–**t4**) or π (all other spectra) pulses were implemented in the CPMG trains[92,93]. 1D $^1$H → $^{119}$Sn CP-CPMG NMR spectra were acquired with multiple $^{119}$Sn transmitter offsets due to the large breadth of the $^{119}$Sn NMR spectra (i.e., VOCS style acquisition; Supplementary Fig. 11)[83,84]. 2D $^1$H → $^{13}$C and $^1$H → $^{119}$Sn CP-HETCOR NMR spectra were recorded with 100 kHz $^1$H RF field of eDUMBO$_{1-22}$ homonuclear dipolar decoupling applied during the $t_1$-evolution period[94]. Each pulse in the homonuclear dipolar decoupling train was 32 µs in duration. 2D $^{119}$Sn adiabatic magic-angle turning (aMAT) NMR spectra were recorded with the published pulse sequences[85–88]. Frequency swept tanh/tan inversion pulses were 100 µs in duration (i.e., 1 rotor-cycle for a 10 kHz MAS frequency) with an *ca.* 90 kHz RF field, a 2 MHz sweep width and 400 points. All $^{119}$Sn aMAT spectra were recorded with $^1$H → $^{119}$Sn CP at the start of the experiment and with the arbitrary indirect dwell (AID) $t_1$ acquisition mode to increase sensitivity[88]. 100 kHz $^1$H RF field of SPINAL-64 heteronuclear decoupling was performed during the acquisition of $^{13}$C and $^{119}$Sn[90].

## Data availability
Raw 1D and 2D NMR data files in Bruker Topspin format are available at https://doi.org/10.5281/zenodo.7569996[95].

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

## Acknowledgements

This work was primarily supported by a grant from Colgate-Palmolive, Inc made to A.J.R. Additional support for solid-state NMR experiments was provided by the National Science Foundation under Grant No. 1709972 to A.J.R. A.J.R. acknowledges additional support from the Alfred P. Sloan Foundation through a Sloan research fellowship. Dynamic nuclear polarization solid-state NMR experiments were performed at the Ames National Laboratory. The Ames National Laboratory is operated for the U.S. DOE by Iowa State University under Contract DE-AC02-07CH11358.

## Author contributions

A.J.R., L.P., Z.H., and C.-y.C. conceived the study and supervised the study. R.W.D. and S.L.C. performed NMR experiments. R.W.D. organized data and wrote the manuscript with contributions from A.J.R. and Z.H.

## Competing interests

C.-y.C., L.P., and Z.H. were or are currently employed by the Colgate-Palmolive Company. The Colgate-Palmolive Company provided financial support for this publication. The remaining authors declare no competing interests.
