## [Peer Review File · Nature Communications]

REVIEWER COMMENTS

Reviewer #1 (Remarks to the Author):

Rossini and Hao, and coworkers have submitted a manuscript that utilizes solution NMR, solid-state NMR (ssNMR), and dynamic nuclear polarization (DNP) enhanced ssNMR spectroscopy to investigate the structure and oxidation state of Tin in toothpaste. It carries industrial significance to identify Sn speciation and oxidation state in toothpaste. However, direct and quantitative commercial product analysis seems technically challenging. The authors have carried out a comprehensive study utilizing chemical shift, chemical shift anisotropy (CSA), and correlation and polarization transfer NMR experiments to identify the coordination state of Sn, and most impressively, quantify Sn(II) and Sn(IV) in model and commercial products. Particularly, DNP has successfully enhanced the NMR sensitivity for ssNMR experiments, enabling efficient analysis and two-dimensional spectroscopy. This study is well-designed for solving industrial problems and, in the meantime, carries good scientific depth and insights. It is an encouraging and inspiring academic and industrial collaboration. I do not have major concerns in terms of the experimental results and analysis or conclusions. However, I would like to provide a few minor comments:

1. The ^{19}F center band assignment in Figure 1B is obtained by comparing it with reference #59. Can the authors run the experiment at a different MAS frequency to confirm?
2. In 2D $^{19}\text{F}\{^{119}\text{Sn}\}$ J-HMQC spectrum of SnF_2 from supplier a, there are Sn(IV) fluoride impurity. How are Sn(IV) peaks assigned? Was it from a reference?
3. The prolonged storage of the DNP samples in a freezer for days resulted in reduced DNP enhancements. The authors rationalize it as reducing the nitroxide biradical caused by converting Sn(II) to Sn(IV). Can this hypothesis be experimentally tested, e.g., by titrating Sn(II) to DNP juice and checking EPR?
4. I would suggest the authors include statements of uncertainty in quantification (Figure 5). The uncertainty can come from a few factors. Is the biradical concentration rigorously 10 mM in all the samples? The authors have mentioned the reduction of the biradical. Is the biradical concentration the same after a series of dissolving and mixing steps for each sample? Do the toothpaste samples have the same concentration? Are the polarization time well-chosen so that all samples have a similar DNP enhancement? Finally, any insights on the quantitative nature of DNP?
5. Can ^{19}F NMR offer quantification?
6. Please indicate the experimental temperature for solution NMR Spectroscopy in Methods.
7. Please indicate the recycle time of F-19 and Sn-119 solution NMR experiments in Methods. Have the T1 relaxation time values for both spins been measured?
8. In Methods, please indicate the ^{19}F decoupling field of room-temperature solid-state NMR Sn-119 experiments.
9. In Methods, please include sample details for solution NMR and room-temperature solid-state NMR experiments. For example, any dilution or filtration of samples for solution NMR? This may have been mentioned in the main text.
10. For DNP samples, has the biradical concentration been optimized? Is there a build-up curve of polarization transfer?
11. For DNP-enhanced ^1H - ^{119}Sn CP experiments, I assume the proton polarization comes from the glassy matrix of the DNP juice. Is there proton-bonded Sn-119?

Additionally, the authors may consider including references for the following statements/topics:

- 1. "The observation of an Sn(IV)-based impurity is important because SnF₂-based toothpastes rely on maximum Sn(II) availability for optimal performance. Impure SnF₂ starting materials will lead to a less effective toothpaste."**
- 2. A review of DNP, e.g., Su et al., Annual Review of Biochemistry, Vol. 84:465-497, 2015**
- 3. Crystal structures of SnF₄ and SnF₂ in Figure 1A.**

Reviewer #2 (Remarks to the Author):

The work of Dorn et al. demonstrates the use of DNP-enhanced solid-state NMR spectroscopy to probe tin oxidation states in commercial toothpastes. The work is extremely thorough, analysing model samples in depth before using the gained knowledge to characterise the toothpastes. The use of DNP was crucial owing to the low wt% of Sn in toothpaste combined with the broad corresponding spectra of Sn(II). As such, I believe the work to be of interest and suitable for publication in Nature Communications after the following points are addressed:

- 1. Line 39, it should state "Conventional solid-state NMR"**
- 2. Line 75, 85% of its tin as Sn(II) – or something to this effect.**
- 3. Line 110, microwave irradiation does not drive the cross-effect transfer (as used in this work), it only saturates electron spin transitions. This should be reworded.**
- 4. Line 128, residing in a symmetric**
- 5. Line 135, NMR spectra**
- 6. Line 150, MAS frequency. An**
- 7. Line 164, if the authors are aware that relativistic effects need to be included, why was this not done, besides the obvious extra computational resources required? The large discrepancy between the calculated CSAs and experimentally-determined values will also accordingly effect the calculated isotropic shifts.**
- 8. Line 165, accurately calculate**
- 9. Line 172, show**
- 10. Line 174, predictions of ¹⁹F chemical shifts are discussed but these are not presented anywhere.**
- 11. Line 175, then what does the ¹¹⁹Sn peak at -750 ppm correspond to, if not SnF₄?**
- 12. Line 212, has ¹⁹F decoupling been shown to be important for this type of Sn system?**
- 13. Line 232, repeat.**
- 14. Line 270, S2). Toothpaste 4...**
- 15. Line 282, this is contradictory. It's stated that enhancements are 12-45 but also similar to 84-105. It should be stated that they cannot be estimated for the ¹¹⁹Sn NMR spectra, but are expected to be the same as for the ¹³C NMR spectra.**
- 16. Line 290, this could also be the OH of glycerol.**
- 17. Line 304, recorded.**
- 18. Line 333, due to differences.**
- 19. Line 376, does this not then indicate that glycerol is predominantly coordinating the tin? This would then explain the large differences with Toothpaste 4, which does not contain glycerol.**
- 20. Line 418, if the above is correct, then this should be edited.**
- 21. Line 475, standard bore?**
- 22. Line 592, reference needs correcting**
- 23. Line 715, reference needs correcting – check all again carefully.**

Response to Reviewers

Reviewer #1 (Remarks to the Author):

Rossini and Hao, and coworkers have submitted a manuscript that utilizes solution NMR, solid-state NMR (ssNMR), and dynamic nuclear polarization (DNP) enhanced ssNMR spectroscopy to investigate the structure and oxidation state of Tin in toothpaste. It carries industrial significance to identify Sn speciation and oxidation state in toothpaste. However, direct and quantitative commercial product analysis seems technically challenging. The authors have carried out a comprehensive study utilizing chemical shift, chemical shift anisotropy (CSA), and correlation and polarization transfer NMR experiments to identify the coordination state of Sn, and most impressively, quantify Sn(II) and Sn(IV) in model and commercial products. Particularly, DNP has successfully enhanced the NMR sensitivity for ssNMR experiments, enabling efficient analysis and two-dimensional spectroscopy. This study is well-designed for solving industrial problems and, in the meantime, carries good scientific depth and insights. It is an encouraging and inspiring academic and industrial collaboration. I do not have major concerns in terms of the experimental results and analysis or conclusions. However, I would like to provide a few minor comments:

1. The ^{19}F center band assignment in Figure 1B is obtained by comparing it with reference #59. Can the authors run the experiment at a different MAS frequency to confirm?

We believe the reviewer is referring to the ^{119}Sn NMR spectrum in Figure 1B. We have recorded ^{119}Sn NMR spectra with 20 kHz and 25 kHz MAS frequencies to confirm the ^{119}Sn isotropic chemical shifts (see below). We have added this Figure to the revised SI as Figure S16 and noted that MAS spectra were recorded with multiple MAS frequencies in the experimental section.

2. In 2D $^{19}\text{F}\{^{119}\text{Sn}\}$ J-HMQC spectrum of SnF_2 from supplier a, there are Sn(IV) fluoride impurity. How are Sn(IV) peaks assigned? Was it from a reference?

We have edited the text to make the justifications for the peak assignments more clear (bottom of page 6):

“On the other hand, the sample from supplier a shows additional ^{19}F and ^{119}Sn NMR signals that are attributed to Sn(IV) fluoride impurities because of the observed ^{19}F chemical shifts and ^{119}Sn chemical shift anisotropy that are similar to those of SnF_4 and correlations observed in the 2D $^{19}\text{F}\{^{119}\text{Sn}\}$ J-HMQC spectrum (Figure 1B, Figure 2 and Figure S2).”

3. The prolonged storage of the DNP samples in a freezer for days resulted in reduced DNP enhancements. The authors rationalize it as reducing the nitroxide biradical caused by converting Sn(II) to Sn(IV). Can this hypothesis be experimentally tested, e.g., by titrating Sn(II) to DNP juice and checking EPR?

We have modified the text to clarify that the loss in DNP enhancements is only observed when SnF_2 is present in the biradical solution. This makes it clear that reaction of nitroxide radicals with Sn(II) is the cause of the lost DNP enhancements. (page 13):

“Prolonged storage of the DNP samples in a freezer for days resulted in no DNP enhancements, likely due to reduction of the nitroxide biradical caused by conversion of Sn(II) to Sn(IV). In the absence of dissolved SnF_2 , the nitroxide biradical water glycerol solutions retain their DNP enhancements after months of storage in a freezer, consistent with reaction of Sn(II) with the nitroxide radicals causing the loss in DNP enhancements.”

4. I would suggest the authors include statements of uncertainty in quantification (Figure 5). The uncertainty can come from a few factors. Is the biradical concentration rigorously 10 mM in all the samples? The authors have mentioned the reduction of the biradical. Is the biradical concentration the same after a series of dissolving and mixing steps for each sample? Do the toothpaste samples have the same concentration? Are the polarization time well-chosen so that all samples have a similar DNP enhancement? Finally, any insights on the quantitative nature of DNP?

The concentration of the biradical is approximately 10 mM. We note that DNP experiments do not require a precise concentration of nitroxide biradicals to obtain high DNP enhancements. It is typical to observe similar DNP enhancements for nitroxide radicals as long as concentration is in the range of 5 to 20 mM (see for example, <https://pubs.rsc.org/en/content/articlelanding/2012/sc/c1sc00550b>). Differences in the different toothpaste densities will cause the concentration across the samples to be slightly different. In summary, variations in the radical concentration may affect the DNP enhancements. But, even if the DNP enhancement is varying from sample to sample it will not affect the quantification of the different Sn species.

The radical concentration will be more or less the same from when the sample was prepared to when experiments were performed as the time from preparation to running experiments was less than 30 minutes. Once the sample is in the magnet, oxidation will be negligible because the sample is completely frozen with a temperature of ca. 110 K.

With regards to Sn quantification, we believe that the discussion on page 21 gives a thorough discussion of the different factors that affect the intensity of the ^{119}Sn NMR signals. The methods and SI also describes optimization of CP contact times and confirms that CPMG acquisition does NOT affect relative peak intensities. As is noted on page 21, if anything we are slightly underestimating the concentration of Sn(II). But, we note that the observed ratios of Sn(II) and Sn(IV) are consistent with prior Sn *K*-edge X-ray spectroscopy studies. Also, we performed the intentional air oxidation experiments where toothpaste was left on the benchtop in air overnight and observed an increase in Sn(IV).

5. Can ^{19}F NMR offer quantification?

When SnF_2 is in solution, F dissociates from Sn. This was observed in the solution ^{19}F NMR spectrum of the model toothpaste. It is anticipated that F is nearly fully dissociated from Sn in the commercial toothpaste as the matrix of the model toothpaste is very similar to commercial toothpaste. Therefore, ^{19}F NMR is not be able to quantify the amount of Sn(II) and Sn(IV) in the toothpaste samples. It could of course possibly be used to confirm that the initial SnF_2 loading in the toothpaste was correct, but there are likely other techniques that are easier.

6. Please indicate the experimental temperature for solution NMR Spectroscopy in Methods.

We have modified the main text and experimental section to indicate that all solution NMR experiments were performed at room temperature.

7. Please indicate the recycle time of ^{19}F and ^{119}Sn solution NMR experiments in Methods. Have the T_1 relaxation time values for both spins been measured?

The recycle delays are given in Table S4. Multiple ^{19}F and ^{119}Sn solution NMR spectra were recorded with different recycle delays to ensure the spectra were recorded quantitatively. We have noted this in the Methods section.

8. In Methods, please indicate the ^{19}F decoupling field of room-temperature solid-state NMR ^{119}Sn experiments.

The ^{19}F decoupling RF field was included in the methods: “SPINAL-64 heteronuclear decoupling with a 50 kHz ^{19}F RF field was performed during the acquisition of ^{119}Sn NMR signals”

9. In Methods, please include sample details for solution NMR and room-temperature solid-state NMR experiments. For example, any dilution or filtration of samples for solution NMR? This may have been mentioned in the main text.

There was no special sample preparation for running NMR experiments besides preparing the samples for DNP which was included in the method.

10. For DNP samples, has the biradical concentration been optimized? Is there a build-up curve of polarization transfer?

The biradical concentration was not optimized. The radical concentration was chosen based on previously reported optimal concentrations of AMUPol in DNP juice. We have noted this in the methods section.

Below is an example of the optimization of the $^1\text{H} \rightarrow ^{119}\text{Sn}$ CP contact time. We have noted in the methods section that a 6 ms contact time was experimentally found to be optimal.

11. For DNP-enhanced ^1H - ^{119}Sn CP experiments, I assume the proton polarization comes from the glassy matrix of the DNP juice. Is there proton-bonded Sn-119?

That is correct. 2D ^1H - ^{119}Sn CP-HETCOR spectra revealed correlations with the same water and glycerol ^1H spins of the matrix that were observed in a 2D ^1H - ^{13}C CP-HETCOR spectrum (see the discussion of Figure 3). In response to a reviewer 2 comment, we noted on page 14:

“The ^{119}Sn DNP enhancements are likely similar to those measured for ^{13}C since in both ^{119}Sn and ^{13}C cross-polarization NMR experiments the magnetization is derived from the same bath of ^1H spins.”

Additionally, the authors may consider including references for the following statements/topics:

1. “The observation of an Sn(IV)-based impurity is important because SnF₂-based toothpastes rely on maximum Sn(II) availability for optimal performance. Impure SnF₂ starting materials will lead to a less effective toothpaste.”

We have added references at the end of the first sentence.

2. A review of DNP, e.g., Su et al., Annual Review of Biochemistry, Vol. 84:465-497, 2015

We have added a citation to this article.

3. Crystal structures of SnF₄ and SnF₂ in Figure 1A.

We have added citations for the crystal structures of SnF₂ and SnF₄.

Reviewer #2 (Remarks to the Author):

The work of Dorn et al. demonstrates the use of DNP-enhanced solid-state NMR spectroscopy to probe tin oxidation states in commercial toothpastes. The work is extremely thorough, analysing model samples in depth before using the gained knowledge to characterise the toothpastes. The use of DNP was crucial owing to the low wt% of Sn in toothpaste combined with the broad corresponding spectra of Sn(II). As such, I believe the work to be of interest and suitable for publication in Nature Communications after the following points are addressed:

We thank the reviewer for their positive comments and for their careful reading of the manuscript. We have addressed their comments below.

1. Line 39, it should state “Conventional solid-state NMR”

Fixed.

2. Line 75, 85% of its tin as Sn(II) – or something to this effect.

Fixed.

3. Line 110, microwave irradiation does not drive the cross-effect transfer (as used in this work), it only saturates electron spin transitions. This should be reworded.

Fixed.

4. Line 128, residing in a symmetric

Fixed.

5. Line 135, NMR spectra

Fixed.

6. Line 150, MAS frequency. An

Fixed.

7. Line 164, if the authors are aware that relativistic effects need to be included, why was this not done, besides the obvious extra computational resources required? The large discrepancy between the calculated CSAs and experimentally-determined values will also accordingly effect the calculated isotropic shifts.

We don't have access to any periodic plane-wave DFT codes that can calculate relativistic chemical shift terms. To the best of our knowledge, there are no plane-wave DFT codes that can do this. Note, that relativistic chemical shift calculations are typically performed with molecular/cluster computational chemistry packages such as ADF, ORCA, ReSpect, etc. However, it is challenging to create relevant structural models of extended ionic solids using molecular computational chemistry packages.

8. Line 165, accurately calculate

Fixed.

9. Line 172, show

Fixed.

10. Line 174, predictions of ^{19}F chemical shifts are discussed but these are not presented anywhere.

We thank the reviewer for pointing out the missing information. We have updated Table S1 to include calculated ^{19}F shielding values. For SnF_2 , isotropic ^{19}F shieldings were calculated to be: 168 ppm, 169 ppm, 180 ppm and 191 ppm. For SnF_4 , isotropic ^{19}F shieldings were calculated to be: 297 ppm and 271 ppm. The predicted differences in isotropic shielding values are consistent with the observed chemical shift differences. We have added text to summarize this (page 8):

“GIPAW calculated ^{19}F chemical shielding values are in reasonable agreement with experiment (Figure S1). The calculations predict that SnF_2 should have ^{19}F chemical shifts which are approximately 100 ppm more positive than those of SnF_4 .”

11. Line 175, then what does the ^{119}Sn peak at -750 ppm correspond to, if not SnF_4 ?

We are not confident on the assignment of the -750 ppm ^{119}Sn NMR signal. However, it is possible that it corresponds to a different SnF_4 polymorph or a hydrated impurity. We have added an additional comment to the main text.

“We suspect that the other unassigned signals are derived from impurities, such as hydrated phases or other polymorphic forms of SnF_4 .”

12. Line 212, has ^{19}F decoupling been shown to be important for this type of Sn system?

We don't definitively know how ^{19}F decoupling would affect the ^{119}Sn NMR signal intensities under DNP conditions. But, we are assuming that any Sn ions with an attached F atom would show reduced homogeneous transverse relaxation time constants (T_2'), similar to how T_2' is reduced if ^1H heteronuclear decoupling is not used during acquisition of solid-state NMR spectra. Because we acquired all NMR spectra with CPMG acquisition, a reduction in T_2' could result in NMR signals that have reduced intensity, possibly even below the limits of detection. We have updated the statement in the text to improve clarity.

“As discussed in more detail below, we assume that only Sn atoms fully dissociated from F will be primarily observed in the DNP-enhanced ^{119}Sn solid-state NMR spectra because we lack the capability to decouple ^{19}F . The sizeable ^{19}F heteronuclear couplings for F coordinated tin ions could lead to reduced ^{119}Sn homogeneous transverse relaxation time constants (T_2') and low ^{119}Sn CPMG NMR signal intensities.”

13. Line 232, repeat.

Fixed.

14. Line 270, S2). Toothpaste 4...

Fixed.

15. Line 282, this is contradictory. It's stated that enhancements are 12-45 but also similar to 84-105. It should be stated that they cannot be estimated for the ^{119}Sn NMR spectra, but are expected to be the same as for the ^{13}C NMR spectra.

We thank the reviewer for their comment. We have revised the text to make these points more clear.

"However, no ^{119}Sn NMR spectrum could be recorded without MW irradiation in a reasonable amount of time (ca. 1 hour). Therefore, we are likely under-estimating the ^{119}Sn DNP enhancements. The ^{119}Sn DNP enhancements are likely similar to those measured for ^{13}C since in both ^{119}Sn and ^{13}C cross-polarization NMR experiments the magnetization is derived from the same bath of ^1H spins."

16. Line 290, this could also be the OH of glycerol.

We have modified the text to note that correlations could also come from the OH of glycerol.

17. Line 304, recorded.

Fixed.

18. Line 333, due to differences.

Fixed.

19. Line 376, does this not then indicate that glycerol is predominantly coordinating the tin? This would then explain the large differences with Toothpaste 4, which does not contain glycerol.

The reviewer raises an interesting point. The fact that the toothpaste 4 does not contain glycerol and shows a ^{119}Sn NMR spectrum distinct from the other toothpastes could indicate that glycerol is the main ligand in the other toothpastes. However, as is noted in the text, toothpaste 4 also contains sorbitol. It is reasonable to assume that sorbitol and glycerol could exhibit similar coordination chemistries given the similarity in their molecular structures. If they do indeed bind to tin ions, both would likely bind with similar affinity. Therefore, it is difficult to add any more definitive statements about what is coordinated to tin.

20. Line 418, if the above is correct, then this should be edited.

See our comments above in response to comment 19.

21. Line 475, standard bore?

Fixed.

22. Line 592, reference needs correcting

Fixed.

23. Line 715, reference needs correcting – check all again carefully.

We fixed this reference and carefully checked all others to make sure complete information is provided.

Structural Characterization of Tin in Toothpaste By Dynamic Nuclear Polarization Enhanced ^{119}Sn Solid-State NMR Spectroscopy

Rick W. Dorn,^{1,2} Scott L. Carnahan,^{1,2} Chi-yuan Chen,³ Long Pan,³ Zhigang Hao,^{3} Aaron J.
Rossini,^{1,2*}*

¹*US Department of Energy Ames National Laboratory, Ames, IA, USA, 50011.*

²*Iowa State University, Department of Chemistry, Ames, IA, USA, 50011.*

³*Colgate-Palmolive Company, Piscataway, NJ, USA 08855.*

AUTHOR INFORMATION

Corresponding Author

*e-mail: zhigang_hao@colpal.com, phone: 732-878-6218.

*e-mail: arossini@iastate.edu, phone: 515-294-8952.

TOC

Abstract

Stannous fluoride (SnF₂) is an effective fluoride source and antimicrobial agent that is widely used in commercial toothpaste formulations. The antimicrobial activity of SnF₂ is partly attributed to the presence of Sn(II) ions. However, it is challenging to directly determine the Sn speciation and oxidation state within commercially available toothpaste products due to the low weight loading of SnF₂ (0.454 wt.% SnF₂, 0.34 wt.% Sn) and the amorphous, semi-solid nature of the toothpaste. Here, we performed dynamic nuclear polarization (DNP) enhanced ¹¹⁹Sn solid-state NMR experiments to directly probe the Sn speciation within commercially available toothpaste. Conventional solid-state NMR spectra of solid Sn(II) fluoride and Sn(IV) fluoride revealed that the ¹⁹F isotropic chemical shift and ¹¹⁹Sn chemical shift anisotropy (CSA) are highly sensitive to the Sn oxidation state; Sn(II) exhibits a near order of magnitude larger CSA than Sn(IV). DNP experiments on model and commercially available toothpastes were enabled by directly mixing the DNP polarization agents (AMUPol biradical) within the toothpaste. The large sensitivity gains provided by DNP enabled the acquisition of 2D ¹¹⁹Sn magic-angle turning (MAT) NMR spectra of all samples. The 2D ¹¹⁹Sn MAT spectra resolve Sn(II) and Sn(IV) species based on the magnitude of their CSA. With knowledge of the ¹¹⁹Sn chemical shift tensor parameters, 1D ¹H→¹¹⁹Sn CP-CPMG NMR spectra were then fit to estimate the populations of Sn(II) and Sn(IV) within the toothpastes. Notably, three of the four commercially available toothpastes contained at least 80 % Sn(II), whereas one of the toothpaste contained a significantly higher amount of Sn(IV).

Introduction

Oral health is highly important for a person's overall physical health and well-being.¹⁻⁴ Toothpaste is the most commonly used dental hygiene product to maintain good oral health by preventing and protecting against oral diseases, such as caries (i.e., cavities/tooth decay) and gingivitis (i.e., gum disease).⁵ Within commercial toothpastes, fluoride is the most effective active ingredient to prevent caries/cavities.^{1, 5-10} Stannous fluoride (SnF_2), one of three fluoride sources recognized by the United State Food and Drug Administration, has been used since the 1950's as an effective way to deliver fluoride ions to tooth enamel and dentin.¹¹⁻¹⁴ However, early toothpastes exhibited challenges associated with the stability of SnF_2 and its compatibility with other ingredients.¹⁴⁻¹⁷ Fortunately, advances in formulation technologies over the last couple decades have resulted in toothpastes with reportedly stable SnF_2 .^{14, 17-21}

The use of SnF_2 as a fluoride source is highly appealing because SnF_2 has been shown to be an effective antimicrobial agent; antimicrobial agents help reduce gingivitis and plaque formation.^{15, 21-30} The antimicrobial properties associated with SnF_2 in toothpaste is thought to arise from the presence of Sn cations with an oxidation state of +2 [Sn(II)].^{21, 23, 31-33} Therefore, it is important that the majority of Sn within commercially available toothpaste maintains the +2 oxidation state.^{21, 23, 31-33} However, Sn(II) cations are generally unstable and will readily oxidize to the more stable Sn(IV) cations upon air exposure. Commercial toothpaste formulations have various additives designed to stabilize Sn(II) compounds within the formulation. But, it is challenging to directly determine the Sn speciation and oxidation state within commercially available toothpaste due to the low weight loading of SnF_2 (0.454 wt.%, 0.34 wt.% Sn) and the amorphous semi-solid nature of the toothpaste. In 2019, Myers and co-workers used Sn K-edge X-ray Absorption Near Edge Spectroscopy (XANES) to suggest that Colgate Total^{SF} contains

approximately 85% of its tin in the Sn(II) oxidation state.¹⁷ More recently, Desmau and co-workers probed the Sn oxidation state within commercially available toothpastes via Sn K-edge XAS.³⁴ In both reports, the relative populations of Sn(II) and Sn(IV) species were estimated by fitting the experimental XAS spectra with XAS spectra of Sn(II) and Sn(IV) standards. However, XAS provides only a partial picture of chemical structure, and energy differences of the Sn(II) and Sn(IV) Sn K-edge spectral features are small (~ 2.5 eV) and overlapping.

Magic-angle spinning (MAS) solid-state NMR spectroscopy is a powerful technique to determine structure within crystalline and amorphous solids. Sn possess three NMR active isotopes (^{115}Sn , ^{117}Sn and ^{119}Sn), with ^{119}Sn generally being the preferred nucleus to probe because it has the largest gyromagnetic ratio (2.7 times lower than ^1H) and natural isotopic abundance (8.59 %). ^{119}Sn exhibits a large isotropic chemical shift range that is highly dependent on the local electronic structure surrounding the Sn atom.³⁵⁻⁴⁵ In addition, the magnitude of the chemical shift anisotropy (CSA) is highly dependent on the symmetry at the Sn atom; asymmetric Sn coordination environments yield large CSA and broad NMR spectra. The magnitude of CSA is often quantified with the span (Ω) which is calculated from the difference of the largest and smallest principal components of the magnetic shielding tensor ($\Omega = \sigma_{33} - \sigma_{11}$) or the chemical shift tensor ($\Omega = \delta_{11} - \delta_{33}$).⁴⁶⁻⁴⁷ In general, Sn(IV) adopts a much more symmetric structure than its corresponding Sn(II) analogs and likely has larger differences in energy between occupied and unoccupied orbitals. For these reasons Sn(IV) compounds tend to have smaller CSA than Sn(II) compounds.^{40, 48-49} For example, ^{119}Sn NMR spectra of Sn(IV) systems typically reveal spans of 0 - 200 ppm,^{36, 40, 48-55} whereas for Sn(II) compounds the span is often between $\sim 700 - 1000$ ppm and can be upwards of ~ 4000 ppm.^{39-41, 48-49, 53, 56-62} We note that there have been a few reports of Sn(IV) or Sn(II) exhibiting relatively large or small spans,

respectively, due to either very distorted or highly symmetric coordination environments, respectively.^{49, 62-65} Unfortunately, conventional room temperature ^{119}Sn MAS NMR spectroscopy of commercial toothpaste is not practically feasible due to the low weight loading of Sn within commercial toothpastes (~ 0.34 wt. % Sn). In addition, the semi-solid nature of toothpaste will likely result in molecular mobility that causes the full or partial averaging of the ^{119}Sn CSA at room temperature. Additionally, solution ^{119}Sn NMR is also hindered by the semi-solid nature of toothpaste.

Here, we apply cryogenic MAS dynamic nuclear polarization (DNP)⁶⁶⁻⁶⁹ to enhance ^{119}Sn solid-state NMR signals of frozen toothpastes by one to two orders of magnitude, allowing 1D ^{119}Sn solid-state NMR spectra to be obtained in minutes from dilute commercial toothpaste formulations. Conventional room temperature solution and solid-state NMR experiments are performed on SnF_2 and SnF_4 . In a MAS DNP experiment, **microwave irradiation is used to saturate electron paramagnetic resonance transitions, resulting** in the **subsequent** transfer of electron spin polarization from stable free radicals to the ^1H spins of the solvent matrix and/or analyte.⁶⁶⁻⁶⁸ Toothpastes typically contain high amounts of water and/or glycerol; water/glycerol mixtures have been shown to be an ideal matrix for MAS DNP experiments.⁶⁶ We note that the use of DNP to enable the acquisition of ^{119}Sn solid-state NMR spectra of dilute Sn(IV) species has been previously demonstrated.^{55, 70-75} To enable DNP experiments the AMUPol biradical⁷⁶ was directly dissolved in toothpaste formulation. The DNP experiments were performed at ca. 110 K on frozen toothpaste. Freezing the toothpaste is beneficial because it eliminates any molecular motion, allowing measurement of ^{119}Sn CSA and enables ^1H - ^{119}Sn cross-polarization (CP) to transfer the DNP-enhanced ^1H polarization to ^{119}Sn nuclei. The ^{119}Sn CSA is shown to be

a sensitive probe of the Sn oxidation state and from fits of the 1D ^{119}Sn solid-state NMR spectra the relative amounts of Sn(II) and Sn(IV) within the formulation can be estimated.

Results and Discussion

Sn(II) and Sn(IV) Fluoride – Chemical Shift and Sample Purity Analysis. We first performed room temperature ^{19}F and ^{119}Sn solid-state NMR spectroscopy on SnF_2 and SnF_4 to determine chemical shift tensor parameters of Sn in the II or IV oxidation state, respectively. SnF_4 features Sn in the more stable IV oxidation state, with each Sn atom residing in a symmetric octahedral environment coordinated by 6 F atoms (Figure 1A).⁷⁷ Consequently, no spinning sidebands are observed in the ^{119}Sn solid-state NMR spectrum of SnF_4 recorded with a 25 kHz MAS frequency, indicating Ω is less than 170 ppm (Figure 1B, upper). Note, throughout this entire manuscript, the Herzfeld-Berger convention is used to report the CSA.⁴⁶⁻⁴⁷ $\alpha\text{-SnF}_2$ contains Sn in the less stable, II oxidation state and there are two unique Sn sites in significantly asymmetric environments that are coordinated by three or five F atoms (Figure 1A).⁷⁸ 1D ^{119}Sn NMR spectra of SnF_2 were recorded for samples purchased from two different suppliers. The ^{119}Sn NMR spectrum of SnF_2 (supplier **b**) reveals two isotropic ^{119}Sn NMR signals with $\delta_{\text{iso}} = -948$ ppm or -1023 ppm and $\Omega = 990$ ppm or 930 ppm, respectively (skew = $\kappa = 1.0$ in both cases; Figure 2B, lower). The ^{119}Sn NMR spectrum of SnF_2 (supplier **b**) is consistent with prior reports and the structure determined from single-crystal X-ray diffraction.^{59, 79} On the other hand, the sample from supplier **a** shows additional ^{19}F and ^{119}Sn NMR signals. These additional NMR signals are attributed to Sn(IV) fluoride impurities because the observed ^{19}F chemical shifts, ^{119}Sn chemical shift anisotropy, and correlations observed in the 2D $^{19}\text{F}\{^{119}\text{Sn}\}$ *J*-HMQC spectrum are similar to those observed for SnF_4 (Figure 1B, Figure 2 and Figure S2). We note

that the ^{19}F chemical shift of tin fluoride materials appear to be sensitive to the Sn oxidation state; SnF_2 and SnF_4 exhibit a $\Delta\delta_{\text{iso}}(^{19}\text{F})$ of 100 ppm (Figure 2A).

Figure 1. (A) Crystal structures of (left) SnF_4 and (right) SnF_2 . (B) 1D ^{119}Sn spin echo NMR spectra of (upper to lower) SnF_4 , SnF_2 from supplier a and SnF_2 from supplier b recorded with a 25 kHz MAS frequency. An analytical simulation for SnF_2 is shown below. (C) 2D $^{19}\text{F}\{^{119}\text{Sn}\}$ J-HMQC NMR spectra of SnF_2 from (upper) supplier a and (lower) supplier b. Asterisks (*) indicate spinning sidebands.

Periodic plane-wave density-functional theory (DFT) calculations utilizing the gauge-including projector-augmented wave (GIPAW) method predicts that the ^{119}Sn isotropic shielding (σ_{iso}) and Ω of two Sn species in SnF_2 differ by 76 and 68 ppm, respectively, and that the most shielded (i.e., most negatively shifted) Sn site exhibits the smaller Ω (Table S1). The most shielded Sn site is coordinated by 5 F atoms. The predicted difference in $\sigma_{\text{iso}}/\delta_{\text{iso}}$ and Ω is in excellent agreement with that observed experimentally, where the δ_{iso} and Ω differ by *ca.* 75 and

60 ppm, respectively. Therefore, we assign the two ^{119}Sn NMR signals with $\delta_{\text{iso}} = -948$ ppm ($\Omega = 990$ ppm) or -1023 ppm ($\Omega = 930$ ppm) to Sn coordinated by three or five F atoms, respectively. We note that while the difference in the DFT calculated Ω is in excellent agreement with that observed experimentally, the magnitude of the DFT calculated Ω is smaller (Table S1). GIPAW calculations do not account for relativistic effects which are likely needed to accurately calculate ^{119}Sn CS tensors.⁸⁰⁻⁸¹ GIPAW calculated ^{19}F chemical shielding values are in reasonable agreement with experiment (Figure S1). The calculations predict that SnF_2 should have ^{19}F chemical shifts which are approximately 100 ppm more positive than those of SnF_4 .

The ^{119}Sn NMR spectrum of SnF_4 shows multiple ^{119}Sn NMR signals that all exhibit small CSA, revealing that this SnF_4 sample contains impurities (Figure 1B, upper). A 1D ^{19}F spin echo NMR spectrum reveals three NMR signals between *ca.* -120 ppm to -160 ppm (Figure 2A); but, only two ^{19}F NMR signals are expected for the terminal and bridging F atoms in a 1:2 ratio, respectively (Figure 1A). We recorded a 2D $^{19}\text{F}\{^{119}\text{Sn}\}$ *J*-HMQC NMR spectrum to better probe the SnF_4 species from the impurities within this material (see Supplementary Material, Figure S1). Only the ^{119}Sn NMR signals at *ca.* -750 ppm and -785 ppm show correlations to two unique ^{19}F NMR signals at *ca.* -130 ppm and -150 ppm; the difference in the ^{19}F isotropic chemical shift [$\Delta\delta_{\text{iso}}(^{19}\text{F})$] for the latter two sites agrees with that predicted by GIPAW DFT calculations ($\Delta\sigma_{\text{iso}}(^{19}\text{F}) = 26$ ppm, Table S1). Therefore, both ^{119}Sn NMR signals at *ca.* -750 ppm and -785 ppm could plausibly be ascribed to SnF_4 . However, we tentatively assign the ^{119}Sn NMR signal -785 ppm to SnF_4 based on the much larger signal intensity observed in the 1D ^{119}Sn spin echo NMR spectrum and the ^{19}F chemical shifts observed in the 1D ^{19}F spin echo NMR spectrum (Figure 1B and S1). We suspect that the other unassigned signals are derived from impurities, such as hydrated phases or other polymorphic forms of SnF_4 .

2D $^{19}\text{F}\{^{119}\text{Sn}\}$ *J*-HMQC NMR spectra of SnF_2 from supplier **b** reveals the expected correlations between all ^{19}F and ^{119}Sn NMR signals associated with SnF_2 (Figure 1C, lower and Figure S3). The 2D $^{19}\text{F}\{^{119}\text{Sn}\}$ *J*-HMQC NMR spectrum of SnF_2 from supplier **a** reveals additional correlations between different ^{19}F and ^{119}Sn NMR signals that were not observed for SnF_2 from supplier **b** (Figure 1C, upper). Notably, two unique sets of ^{19}F NMR signals of SnF_2 from supplier **a** resonate at significantly lower $\delta_{\text{iso}}(^{19}\text{F})$ than expected for SnF_2 ; the $\delta_{\text{iso}}(^{19}\text{F})$ is near that observed for SnF_4 (Figure S2). The lowest frequency ^{19}F NMR signal correlates with a ^{119}Sn that exhibits no observable CSA with a 25 kHz MAS frequency. Based on the low frequency ^{19}F NMR signal and small ^{119}Sn CSA, we assign these NMR signals to a Sn(IV) fluoride impurities. We note that the observed ^{19}F and ^{119}Sn NMR signals are different than those assigned to SnF_4 .

In summary, SnF_2 exhibits a significantly larger ^{119}Sn CSA than SnF_4 , illustrating that the ^{119}Sn CSA is a good probe of the Sn oxidation state, consistent with prior literature. Additionally, the ^{19}F chemical shifts for SnF_2 were much more positive than those for SnF_4 , suggesting that ^{19}F NMR can also provide insight into the Sn oxidation state. Interestingly, impurities in the SnF_2 and SnF_4 samples from supplier **a** were detected in 1D ^{19}F and ^{119}Sn NMR spectra. Notably, a 2D $^{19}\text{F}\{^{119}\text{Sn}\}$ *J*-HMQC spectrum of SnF_2 from supplier **a** revealed an Sn(IV) fluoride impurity. The observation of an Sn(IV)-based impurity is important because SnF_2 -based toothpastes rely on maximum Sn(II) availability for optimal performance. Impure SnF_2 starting materials will lead to a less effective toothpaste. ^{19}F and ^{119}Sn MAS solid-state NMR spectroscopies are good tools to determine the purity of tin fluoride materials which may be used as precursors in toothpaste products.

Solution ^{19}F and ^{119}Sn NMR Spectroscopy. We performed room temperature solution ^{19}F and ^{119}Sn NMR spectroscopy experiments on a model toothpaste to initially probe all Sn and F atoms before studying commercially available toothpastes with DNP-enhanced ^{119}Sn NMR spectroscopy. The model toothpaste consisted of *ca.* 2 wt. % SnF_2 in a 1:2 mixture of D_2O :glycerol_{d-8}; the solution was prepared *ca.* 1 month before running NMR experiments. The use of only SnF_2 in D_2O :glycerol_{d-8} makes this a simplified version of most SnF_2 -based toothpastes, allowing for an easier assessment of the Sn speciation before studying more complex commercial toothpaste formulations. We note that the solution ^{19}F and ^{119}Sn NMR experiments suggest that the majority of F atoms are dissociated from Sn. As discussed in more detail below, we assume that only Sn atoms fully dissociated from F will be primarily observed in the DNP-enhanced ^{119}Sn solid-state NMR spectra because we lack the capability to decouple ^{19}F . The sizeable ^{19}F heteronuclear couplings for F coordinated tin ions could lead to reduced ^{119}Sn homogeneous transverse relaxation time constants (T_2') and low ^{119}Sn CPMG NMR signal intensities.

A 1D ^{19}F solution NMR spectrum of the model toothpaste reveals primarily two ^{19}F NMR signals at -88.3 ppm and -157.4 ppm (Figure 2A). The broad hump from -150 ppm to -200 ppm are probe background ^{19}F NMR signals (Figure S4). Closer examination of the -157.4 ppm ^{19}F NMR signal reveals three set of doublets centered around the isotropic NMR signal which correspond to 1-bond J -couplings of ^{19}F - ^{119}Sn (1571 Hz), ^{19}F - ^{117}Sn (1505 Hz) and ^{19}F - ^{115}Sn (1385 Hz) (1J ; Figure 2C, upper and Figure S4). The integral of each doublet matches with the corresponding Sn isotopic abundances and the J -couplings scale with the gyromagnetic ratios of the tin isotopes. We recorded a 2D $^{19}\text{F}\{^{119}\text{Sn}\}$ J -HMQC solution NMR spectrum to probe all F atoms bonded to Sn (Figure S5). Only the ^{19}F NMR signals near -155 ppm were observed in the

2D J -HMQC NMR spectrum. Therefore, we assign the ^{19}F NMR signal at -90 ppm to free fluoride ions. Integration of the ^{19}F NMR signals reveals that at least *ca.* 92 % of F is present as ions within the $\text{D}_2\text{O}/\text{glycerol}_{d-8}$ mixture (Figure 2A). We note that the population of solvated F ions is likely even higher as the ^{19}F transmitter was on resonance with the F-Sn ^{19}F NMR signal and due to overlap of the -157.4 ppm signal with the probe background ^{19}F NMR signals (see *Methods*).

Figure 2. (A) ^{19}F spin echo NMR spectra of (upper to lower) the model toothpaste, SnF_4 and SnF_2 (supplier **b**) recorded in either (upper) solution or (lower two) the solid-state. Asterisks indicate spinning sidebands. (B) Modified $^{19}\text{F}\{^{119}\text{Sn}\}$ J -HSQC pulse sequence that allows for evolution of the ^{19}F - ^{119}Sn J -coupling during the ^{119}Sn t_1 -evolution period. (C) Solution 2D $^{19}\text{F}\{^{119}\text{Sn}\}$ J -HSQC NMR spectrum of the model toothpaste acquired with the pulse sequence shown in (B).

The ^{19}F NMR signal assigned to an F-Sn species resonates at a similar shift as was observed for SnF_4 (Figure 2A). To confirm the origin of the -157.4 ppm ^{19}F NMR signal we recorded a 2D $^{19}\text{F}\{^{119}\text{Sn}\}$ J -HSQC solution NMR spectrum using the modified pulse sequence shown in Figure 2B. The modified pulse sequence does not have a central refocusing (π) pulse

applied to the ^{19}F spins during ^{119}Sn t_1 evolution, which causes evolution of ^{19}F - ^{119}Sn J -couplings and ^{119}Sn chemical shifts in the indirect dimension, resulting in a multiplet dependent on the number of attached F atoms to Sn. The modified 2D $^{19}\text{F}\{^{119}\text{Sn}\}$ J -HSQC solution NMR spectrum reveals a septet like pattern in the ^{119}Sn dimension and a doublet in the ^{19}F dimension (^{119}Sn decoupling was not performed during acquisition, Figure 2C). Note, the central line of the septet pattern is absent because the central line of the multiplet arises from ^{119}Sn spins coupled to 3 ^{19}F spin-up and 3 ^{19}F spin-down leading to an effective J -coupling of 0 Hz. The observed ^{119}Sn septet matches exactly to that of a numerical simulation for Sn attached to six F atoms; the three doublets exhibit splittings at 2, 4 or 6 times $^1J_{\text{Sn-F}}$ and the intensities are consistent with an AX_6 spin system determined from a modified Pascal triangle.⁸² Therefore, the ^{19}F NMR signal at -157.4 ppm is assigned to the $(\text{SnF}_6)^{-2}$ anions.

In summary, integration of the 1D ^{19}F solution NMR spectrum suggests at most *ca.* 8 % of F is directly bonded to Sn (Figure 2A). However, assuming that the two ^{19}F NMR signals correspond to free fluoride anions (F^-) and $(\text{SnF}_6)^{-2}$, then only *ca.* 2 % of Sn within the model toothpaste contains F bonds, consistent with a 1D ^{119}Sn solution NMR spectrum that does not show appreciable amounts of Sn-F species (Figure S6). This observation is important because it implies essentially all Sn atoms are observable in the DNP ^{119}Sn solid-state NMR experiments. Sn species exhibiting F bonds would exhibit significant ^{119}Sn NMR signal attenuation in DNP solid-state NMR due to large ^{19}F - ^{119}Sn dipolar couplings that will not be effectively averaged in the absence of ^{19}F heteronuclear decoupling.

Dynamic Nuclear Polarization ^{119}Sn Solid-State NMR Spectroscopy. We performed dynamic nuclear polarization (DNP) enhanced ^{119}Sn solid-state NMR spectroscopy on the model toothpaste, a commercially available preventative gel (preventative gel **1**) and four commercially

available toothpastes (toothpastes **1-4**). The weight loadings of SnF₂ in the model toothpaste, preventative gel **1** and toothpastes **1-4** are *ca.* 2 wt.%, 0.40 wt.% or 0.454 wt.%, corresponding to absolute Sn loadings of *ca.* 1.5 wt.%, 0.30 wt.% or 0.34 wt.%, respectively. Preventative gel **1** contains primarily glycerol, similar to the model toothpaste (glycerol and water, Table S2). Toothpastes **1** and **3** are glycerol-based, while toothpastes **2** and **4** are water and glycerol and/or sorbitol-based. In addition the toothpastes contain other ingredients, such as abrasives (Table S2). Toothpaste **4** included SnCl₂ as an SnF₂ stabilizer. Samples were prepared for DNP experiments by directly dissolving the AMUPol biradical at a final concentration of 11 mM within the toothpastes (see *Methods*). For the model toothpaste, the sample was slightly diluted with H₂O (10:30:60 ratio of H₂O:D₂O:glycerol_{d-8}) to increase DNP enhancements. All DNP experiments were performed immediately after biradical addition. Prolonged storage of the DNP samples in a -20 °C freezer for days resulted in no DNP enhancements, likely due to reduction of the nitroxide biradical caused by conversion of Sn(II) to Sn(IV). In the absence of dissolved SnF₂, the nitroxide biradical water glycerol solutions retain their DNP enhancements after months of storage in a freezer, consistent with reaction of Sn(II) with the nitroxide radicals causing the loss in DNP enhancements.

¹H→¹³C cross-polarization magic-angle spinning (CPMAS) DNP enhancements (ϵ) of the model toothpaste and preventative gel **1** were *ca.* 84 and 105, respectively (Figure 3A and Figure S7). This means that acquisition of solid-state NMR spectra with the same signal-to-noise (SNR) ratio would take more than a thousand times longer to acquire without MW irradiation (i.e., no DNP). The DNP enhancement (ϵ) measured with ¹H→¹¹⁹Sn CP-CPMG experiments was estimated to be between 12 – 45 for the model toothpaste and toothpastes **1-4** (Figure 3B and Figure S8). However, a ¹¹⁹Sn NMR spectrum could not be recorded without MW irradiation in a

reasonable amount of time (*ca.* 1 hour). Therefore, we are likely under-estimating the ^{119}Sn DNP enhancements. The ^{119}Sn DNP enhancements are likely similar to those measured for ^{13}C since in both ^{119}Sn and ^{13}C cross-polarization NMR experiments the magnetization is derived from the same bath of ^1H spins. The high DNP enhancements enables the acquisition of 2D ^1H - ^{119}Sn CP heteronuclear correlation (HETCOR) and magic-angle turning (MAT) NMR spectra of all samples.

A 2D ^1H → ^{13}C CP-HETCOR NMR spectrum of the model toothpaste reveals correlations between the -OH ^1H NMR signals of glycerol or H_2O and the ^{13}C NMR signals of glycerol (Figure 3C). The glycerol CH_x ^1H NMR signals were not observed as the glycerol was fully deuterated. A 2D ^1H → ^{13}C CP-HETCOR NMR spectrum of preventative gel 1 reveals the expected correlations between all the ^1H and ^{13}C NMR signals of the CH_x and -OH groups of the glycerol solvent (Figure S9A). Interestingly, 2D ^1H → ^{119}Sn CP-CPMG HETCOR NMR spectra of the model toothpaste and preventative gel 1 both display correlations between all ^1H NMR signals observed in the 2D ^1H → ^{13}C CP-HETCOR NMR spectra with a broad ^{119}Sn NMR signal (Figure 3D and Figure S9B). The observed ^1H - ^{119}Sn correlations reveal that the Sn is present as ions that are likely solvated by H_2O and/or glycerol. The interaction between Sn(II) and glycerol was also observed in liquid chromatography mass spectrometry (LCMS) experiments (Figure S10).

Figure 3. (A-B) Comparison of (A) $^1\text{H} \rightarrow ^{13}\text{C}$ CPMAS and (B) $^1\text{H} \rightarrow ^{119}\text{Sn}$ CP-CPMG NMR spectra of the model toothpaste recorded (black) with or (red) without microwave (MW) irradiation. The DNP enhancements (ϵ) are given in the figure. DNP-enhanced 2D (C) $^1\text{H} \rightarrow ^{13}\text{C}$ and (D) $^1\text{H} \rightarrow ^{119}\text{Sn}$ CP-HETCOR NMR spectra of the model toothpaste were recorded with a 10 kHz MAS frequency, eDUMBO₁₋₂₂ ^1H homonuclear decoupling during ^1H indirect dimension evolution time, and CPMG detection of ^{119}Sn .

1D $^1\text{H} \rightarrow ^{119}\text{Sn}$ CP-CPMG NMR spectra of all samples were acquired with multiple ^{119}Sn transmitter offsets (VOCS acquisition⁸³⁻⁸⁴) due to the large breadth of the ^{119}Sn NMR signals and relatively low ^{119}Sn NMR excitation bandwidth (Figure S11). The 1D $^1\text{H} \rightarrow ^{119}\text{Sn}$ CP-CPMG NMR spectra of the model toothpaste and preventative gel **1** reveal primarily a broad ^{119}Sn NMR signal *ca.* 1500 ppm in breadth, a larger range than was observed for SnF_2 (*ca.* 1000 ppm).

Likewise, the 1D $^1\text{H} \rightarrow ^{119}\text{Sn}$ CP-CPMG NMR spectra of toothpastes **1-4** reveal broad ^{119}Sn NMR signals that cover a range of *ca.* 1000-1500 ppm in addition to sharper ^{119}Sn NMR features. The broad ^{119}Sn NMR signals likely correspond to Sn(II) species whereas the sharper features likely correspond to Sn(IV). However, identification of Sn(II) and Sn(IV) species is fairly ambiguous from the 1D NMR spectra alone. The 1D ^{119}Sn MAS solid-state NMR spectra are likely broad and featureless because of the presence of isotropic chemical shift distributions, which are typical of disordered and amorphous systems. There is probably a distribution in the number of water, hydroxide, and glycerol (or glyceroxide) molecules coordinated to each tin ion.

To resolve Sn(II) from Sn(IV) species based on their CSA, we recorded DNP-enhanced 2D ^{119}Sn adiabatic magic angle turning (aMAT) NMR spectra of all samples (Figure 4 and S12-S15).⁸⁵⁻⁸⁸ These 2D NMR experiments required only *ca.* 6 hours for the model toothpaste (Sn loading \sim 1.5 wt. %) and *ca.* 16-17 hours for preventative gel **1** and toothpastes **1-4** (Sn loading = 0.3 or 0.34 wt. %, respectively). In a MAT NMR experiment, an isotropic NMR spectrum free of spinning sidebands (indirect dimension) is correlated with its corresponding anisotropic MAS NMR spectrum (direct dimension). Therefore, the anisotropic NMR spectra extracted at specific isotropic chemical shifts (δ_{iso}) can be easily fit to determine the span (Ω) and skew (κ) because the δ_{iso} is known.

2D ^{119}Sn aMAT NMR spectra of the model toothpaste and preventative gel **1** are near identical and reveal broad isotropic ^{119}Sn NMR spectra in the region of *ca.* -600 to -1000 ppm (Figure 4A and S12A). The broad isotropic ^{119}Sn NMR spectra illustrate that there are large distributions in the ^{119}Sn δ_{iso} , likely due to differences in the Sn ion coordination from the solvent matrix. We note that the center of the isotropic ^{119}Sn NMR spectrum of the model toothpaste appears at the same ^{119}Sn chemical shift observed in solution, however, the breadth of the signal

is much narrower at room temperature due to dynamics of the Sn ions in solution (Figure S6). Anisotropic ^{119}Sn NMR spectra extracted from the isotropic ^{119}Sn dimension reveals primarily Sn sites with a Ω of *ca.* 1200 ppm and a κ of +0.7, which are assigned to Sn(II) species based on the large CSA (Figure 4B and S12B). However, anisotropic ^{119}Sn NMR spectra extracted at higher ^{119}Sn δ_{iso} of *ca.* -750 ppm or -800 ppm to -600 ppm for the model toothpaste or preventative gel **1**, respectively, show an increased intensity for the isotropic (centreband) NMR signals. An increase in intensity for the isotropic NMR signal reveals there are additional sites with spans of 150 ppm or less, but with identical ^{119}Sn isotropic chemical shift to sites that have a larger span; the small CSA sites should correspond to Sn(IV). Therefore, the 2D ^{119}Sn aMAT NMR spectra clearly reveal both Sn(II) and Sn(IV) sites based on their CSA. We note that the experimental anisotropic NMR spectra show distorted signal intensities at lower ^{119}Sn chemical shifts due to limited ^{119}Sn NMR excitation bandwidth. Nevertheless, Ω and κ can still be accurately determined by fitting the most intense spinning sidebands with the isotropic shift as a fixed constraint.

Figure 4. (A, C) DNP-enhanced 2D ^{119}Sn aMAT NMR spectra of (A) the model toothpaste and (C) toothpaste 1 acquired with a 10 kHz MAS frequency, $^1\text{H} \rightarrow ^{119}\text{Sn}$ CP at the start of the experiment, and CPMG for ^{119}Sn detection. (B, D) ^{119}Sn anisotropic NMR spectra extracted from the 2D aMAT NMR spectra at the indicated ^{119}Sn isotropic chemical shifts (δ_{iso}). Analytically simulated spectra are shown (colored) below the (black) experimental MAS spectra.

DNP-enhanced 2D ^{119}Sn aMAT NMR spectra of toothpastes **1-3** are near identical and display three relatively sharp isotropic ^{119}Sn NMR signals at *ca.* -762 to -775 ppm, -732 to -744 ppm and -665 ppm, in addition to a broad isotropic ^{119}Sn NMR signal from *ca.* -500 to -600 ppm (Figure 4C, S13A and S14A). The relatively sharp ^{119}Sn NMR signals at *ca.* -762 to -775 ppm and -732 to -744 ppm clearly show intense isotropic ^{119}Sn NMR signals, which are assigned to Sn(IV) species based on the small Ω (*ca.* 150 ppm; Figure 4A, S13B and S14B). There are additional weak sidebands associated with Sn that have spans on the order of *ca.* 1200 ppm, suggesting some Sn(II) species are present at these isotropic shifts. Anisotropic ^{119}Sn NMR spectra extracted at more positive ^{119}Sn δ_{iso} of *ca.* -665 ppm and -600 ppm reveal significantly more intense broad ^{119}Sn NMR spectra with spans of *ca.* 1200 ppm (Figure 4A, S13B and S14B). At the lower ^{119}Sn δ_{iso} of *ca.* -665 ppm, the isotropic NMR signal has increased signal intensity, consistent with additional small CSA sites ($\Omega \approx 150\text{-}250$ ppm). However, the more positively shifted isotropic ^{119}Sn NMR signals are clearly primarily associated with Sn(II) sites. We note that the isotropic ^{119}Sn NMR spectra are not representative of the Sn(II) and Sn(IV) populations due to differences in MAT efficiencies for high or low CSA sites, respectively. The similarities in the 2D ^{119}Sn aMAT spectra of toothpastes **1** and **3** are not surprising because they are both primarily glycerol-based. However, the similarities between the MAT NMR spectrum of toothpaste **2** with the MAT NMR spectra of toothpastes **1** and **3** are interesting because toothpaste **2** contains a significant amount of water in addition to glycerol.

Interestingly, the 2D ^{119}Sn aMAT NMR spectrum of toothpaste **4** is significantly different from that of toothpastes **1-3** (Figure S15A). As mentioned above, toothpaste **4** contains primarily water and sorbitol (similar to toothpaste **2**), in addition to SnCl_2 as a SnF_2 stabilizer (Table S2). The isotropic ^{119}Sn NMR spectrum of toothpaste **4** shows primarily three isotropic ^{119}Sn NMR

signals at *ca.* -665 ppm, -550 ppm and -475 ppm (Figure S15B). Similar ^{119}Sn isotropic NMR signals were observed for toothpaste **1-3**, however, for toothpaste **4** the ^{119}Sn isotropic NMR signals at *ca.* -665 and -550 ppm correspond to predominantly small CSA sites ($\Omega \sim 150\text{-}220$ ppm; Figure S15B). The ^{119}Sn isotropic NMR signal at *ca.* -475 ppm clearly shows sites with both large and small CSA ($\Omega \approx 220$ ppm and 1100 ppm).

With knowledge of all ^{119}Sn chemical shift tensors, the 1D $^1\text{H} \rightarrow ^{119}\text{Sn}$ CP-CPMG NMR spectra of all samples could be analytically simulated (Figure 5 and Table S3). The 2D ^{119}Sn aMAT NMR spectra revealed large distributions in the ^{119}Sn δ_{iso} . Therefore, we fit the 1D $^1\text{H} \rightarrow ^{119}\text{Sn}$ CP-CPMG NMR spectra to sites containing large amounts of Gaussian line broadening to represent the distributions in the isotropic chemical shifts.

The model toothpaste and preventative gel **1** display similar 1D $^1\text{H} \rightarrow ^{119}\text{Sn}$ CP-CPMG NMR spectra, where the populations of Sn(II) were determined to be *ca.* 92 or 93 %, respectively (Figure 5). Toothpastes **1-3** also display similar 1D $^1\text{H} \rightarrow ^{119}\text{Sn}$ CP-CPMG NMR spectra (Figure 5). The populations of Sn(II) were determined to be *ca.* 80 % for toothpastes **1** and **3** and 90 % for toothpaste **2**. We also recorded a 1D $^1\text{H} \rightarrow ^{119}\text{Sn}$ CP-CPMG NMR spectrum of toothpaste **2** after allowing the toothpaste to dry out and exposing to air over the course of 1 day (Figure 5). Interestingly, the population of Sn(II) decreased from *ca.* 90 to 83 %, resulting from oxidation of Sn(II) to Sn(IV) from O_2 in the atmosphere. This experiment was an important control that confirms our hypothesis that the ^{119}Sn NMR signals of Sn(IV) primarily have small spans, while those associated with Sn(II) have larger spans. Consistent with the differences observed in the 2D ^{119}Sn aMAT NMR spectrum, toothpaste **4** displays a different 1D $^1\text{H} \rightarrow ^{119}\text{Sn}$ CP-CPMG NMR spectrum with a significantly lower amount of Sn(II) (*ca.* 68 %; Figure 5). We

note that the relative populations of Sn(II) for toothpastes **1-4** determined here are generally consistent with prior measurements made in Sn K-edge X-ray absorption studies.^{17, 34}

Figure 5. DNP-enhanced $^1\text{H} \rightarrow ^{119}\text{Sn}$ CP-CPMG NMR spectra of the (upper to lower) model toothpaste, preventative gel **1**, and toothpastes **1 – 4**. Multiple CP-CPMG NMR spectra were recorded with different ^{119}Sn transmitter frequencies (i.e., VOCS style acquisition, Figure S11). All spectra were recorded with a 10 kHz MAS frequency and a sample temperature of ~ 100 K.

There are three main mechanisms that can lead to inaccurate Sn(II) and Sn(IV) populations determined from the 1D $^1\text{H} \rightarrow ^{119}\text{Sn}$ CP-CPMG NMR spectra: (1) differences in DNP

enhancement, (2) difference in $^1\text{H}\rightarrow^{119}\text{Sn}$ CP dynamics, and (3) differences in ^{119}Sn refocused transverse relaxation time constants (T_2'). DNP enhancements for Sn(II) and Sn(IV) should be identical since 2D $^1\text{H}\rightarrow^{119}\text{Sn}$ CP-HETCOR NMR spectra revealed that Sn is present as ions within the solvent matrix and ^1H spin diffusion should distribute the DNP enhanced ^1H polarization homogeneously across the frozen solution (Figure 3D and Figure S9B). $^1\text{H}\rightarrow^{119}\text{Sn}$ CP dynamics are likely similar for Sn(II) and Sn(IV) sites since they are both likely coordinated by water, hydroxide ions and/or glycerol molecules. However, $^1\text{H}\rightarrow^{119}\text{Sn}$ CP is likely less efficient for sites with high CSA because the CSA is comparable to or larger than the RF field used for the ^{119}Sn spin-lock pulse. From this perspective, the Sn(II) populations are likely a lower bound. To assess differences in ^{119}Sn T_2' , we investigated the effect that the number of CPMG echoes used during that acquisition of $^1\text{H}\rightarrow^{119}\text{Sn}$ CP-CPMG NMR spectra of the model toothpaste and toothpaste **1** had on the determined Sn(II) and Sn(IV) populations (Figure S16). $^1\text{H}\rightarrow^{119}\text{Sn}$ CP-CPMG NMR spectra processed with 1-100 CPMG echo trains reveal near identical populations of Sn(II) and Sn(IV), confirming that the ^{119}Sn T_2' must be similar for all species. The observation of similar ^{119}Sn T_2' for all Sn species is also consistent with minimal Sn sites exhibiting F bonds, as those sites would exhibit a shorter ^{119}Sn T_2' . Therefore, analytical simulations of the $^1\text{H}\rightarrow^{119}\text{Sn}$ CP-CPMG NMR spectra likely reveal relatively accurate populations of Sn(II) and Sn(IV), where the population of Sn(II) should be taken as a lower bound due to differences in CP efficiencies.

Conclusion

In this contribution, we applied dynamic nuclear polarization (DNP) enhanced ^{119}Sn solid-state NMR spectroscopy to determine the Sn oxidation state and speciation within

commercially available SnF₂-based toothpastes that contain loadings of less than 0.5 wt. %. We first obtained room-temperature ¹⁹F and ¹¹⁹Sn solid-state NMR spectra of SnF₂ and SnF₄. These experiments confirmed Sn(II) exhibits a near order of magnitude larger span than that of Sn(IV), consistent with prior literature. NMR studies of SnF₂ purchased from two different suppliers revealed a significant amount of Sn and F-based impurities in one of the samples. Notably, 2D ¹⁹F {¹¹⁹Sn} *J*-HMQC NMR spectra revealed that the impure SnF₂ sample contains Sn(IV) fluoride-based impurities. ¹⁹F and ¹¹⁹Sn solid-state NMR are good probes of the purity of SnF₂ precursors used in the production of toothpastes. Solution ¹⁹F and ¹¹⁹Sn NMR studies on a model toothpaste consisting of *ca.* 2 wt. % SnF₂ in a 50:50 mixture of D₂O:glycerol_{d-8} suggested that only *ca.* 2 % of the dissolved Sn ions contain F bonds. DNP experiments of model and commercially available toothpastes were enabled by directly mixing the DNP polarizing agent (AMUPol biradical) within the toothpaste. Importantly, the sensitivity gains offered by DNP enabled detection of ¹¹⁹Sn NMR signals from toothpastes with loadings of 0.34 wt.% Sn. 2D ¹H→¹³C and ¹H→¹¹⁹Sn CP-HETCOR NMR spectra of the model toothpaste and preventative gel **1** suggested that all Sn is present as ions that are solvated by water, hydroxide anions and/or glycerol. 1D ¹H→¹¹⁹Sn CP-CPMG NMR spectra of all toothpastes revealed broad ¹¹⁹Sn NMR spectra, with some additional sharper features. Acquisition of 2D ¹¹⁹Sn magic-angle turning (MAT) NMR spectra of all samples allowed for the unambiguous identification of Sn(II) and Sn(IV) species based on their CSA. With knowledge of the ¹¹⁹Sn chemical shift tensor parameters, 1D ¹H→¹¹⁹Sn CP-CPMG NMR spectra were fit to estimate the populations of Sn(II) and Sn(IV) within the toothpastes. Notably, three of the four commercially available toothpastes contained at least 80 % Sn(II), whereas one of the toothpaste contained a significantly higher amount of Sn(IV).

We have demonstrated that DNP-enhanced ^{119}Sn solid-state NMR spectroscopy is an ideal technique to probe the Sn speciation with commercially available toothpastes. The determination of the Sn(II) and Sn(IV) populations within commercially available toothpastes is important both to assess the quality of current formulations and to develop new and improved formulations. Increasing the amount of Sn(II) should increase the antimicrobial properties of SnF_2 -based toothpastes. We observed that both glycerol-based toothpastes (**1** and **3**) and toothpastes containing high amounts of water and glycerol (**3**) can exhibit high amounts of Sn(II) (*ca.* 80-90 %). However, the Sn(II) is readily oxidized to Sn(IV) after prolonged air-exposure. More detailed studies on the specific coordination of Sn and their interactions with other common toothpaste ingredients are on-going in our labs. By better understanding how common toothpaste ingredients interact with Sn, and specifically Sn(II), DNP-enhanced ^{119}Sn solid-state NMR spectroscopy will enable the rational design and development of next-generation SnF_2 -based toothpastes that exhibit increased Sn(II) availability and long-term oxidation stability.

Methods

Solution NMR Spectroscopy. Room temperature solution ^{19}F and ^{119}Sn NMR experiments were performed on the model toothpaste and recorded on a 9.4 T ($\nu_0(^1\text{H}) = 400$ MHz) Bruker standard-bore magnet equipped with a AVANCE NEO console and a liquid- N_2 cooled Bruker Prodigy HXY NMR probe. ^{19}F and ^{119}Sn chemical shifts were referenced to CCl_3F and SnMe_4 , respectively, with D_2O as the lock signal. The ^{19}F $\pi/2$ and π pulses were 15 and 30 μs in duration, corresponding to a 16.7 kHz radio frequency (RF) field. We note that the ^{19}F NMR spectrum was recorded with the ^{19}F transmitter on resonance with the SnF_6^{2-} ^{19}F NMR signal. The ^{19}F NMR signals of the free F ions were *ca.* 26 kHz away from the ^{19}F transmitter. The ^{19}F spin echo NMR spectrum was recorded with 100 μs delays on each side of the ^{19}F π pulse. The ^{19}F solution NMR spectra were acquired with different recycle delays to ensure that quantitative relative peak intensities were obtained. Recycle delays used for solution NMR experiments are given in Table S4. The ^{119}Sn $\pi/2$ and π pulses were 12.5 and 25 μs in duration, corresponding to an *ca.* 20 kHz RF field. The ^{119}Sn spin echo NMR spectrum was recorded with 10 μs delays on each side of the ^{119}Sn π pulse.

Room Temperature Solid-State NMR Spectroscopy. Solid-State NMR spectroscopy experiments were performed at room temperature on a 9.4 T ($\nu_0(^1\text{H}) = 400$ MHz) Bruker wide-bore magnet equipped with a Bruker AVANCE III HD console and a 2.5 mm HXY magic-angle spinning (MAS) NMR probe configured in triple resonance mode. We note that the ^{19}F and ^{119}Sn match was relatively poor (*ca.* 30 and 60 % for ^{19}F and ^{119}Sn , respectively) when tuned simultaneously to ^{19}F and ^{119}Sn on the ^1H and X channel, respectively. The magnetic field strength was referenced to 1 % tetramethyl silane (TMS) in CDCl_3 with adamantane ($\delta(^1\text{H}) = 1.71$ ppm) as a secondary chemical shift reference. ^{19}F and ^{119}Sn chemical shifts were indirectly

referenced to CCl_3F and SnMe_4 , respectively, using the previously published IUPAC recommended relative NMR frequencies.⁸⁹

The ^{19}F $\pi/2$ and π pulses were 4 and 8 μs in duration, corresponding to a 62.5 kHz radio frequency (RF) field. The ^{119}Sn $\pi/2$ and π pulses were 3.5 and 7 μs in duration, corresponding to a 71 kHz RF field. 2D $^{19}\text{F}\{^{119}\text{Sn}\}$ J -based heteronuclear multiple quantum correlation (J -HMQC) experiments were recorded with our previously described arbitrary indirect dwell (AID) HMQC sequences.⁸⁸ SPINAL-64 heteronuclear decoupling with a 50 kHz ^{19}F RF field was performed during the acquisition of ^{119}Sn NMR signals.⁹⁰ ^{119}Sn and ^{19}F solid-state NMR spectra were acquired at multiple MAS frequencies to confirm the assignment of isotropic and sideband NMR signals (Figure S17).

Dynamic-Nuclear Polarization-Enhanced Solid-State NMR Spectroscopy. Dynamic-nuclear polarization (DNP) enhanced solid-state NMR spectroscopy experiments were performed on a 9.4 T ($\nu_0(^1\text{H}) = 400$ MHz) Bruker wide-bore magnet equipped with a 263 GHz gyrotron, a Bruker AVANCE III console and a 3.2 mm HXY MAS DNP NMR probe. The samples were cooled to a temperature of *ca.* 100 K with a Bruker liquid- N_2 cooling cabinet. NMR experiments were performed with the probe in either HXY triple or HX double resonance mode. DNP experiments were enabled by directly mixing the AMUPol biradical within the toothpastes at a concentration of *ca.* 10 mM, since this radical concentration has been reported to give the highest DNP enhancements and sensitivity gains.⁶⁶ A typical sample preparation of the commercial toothpastes for DNP consisted of weighing out the AMUPol biradical in a vial (*ca.* 1.3 – 2.3 mg), adding the proper amount of toothpaste to reach a concentration of *ca.* 10 mM, and then vigorously stirring the toothpaste for *ca.* 10 min to ensure the radical was homogeneously mixed throughout the toothpaste. The densities of the toothpastes were assumed

to be *ca.* 1.3 g cm⁻³. We note that samples of toothpastes **1** – **4** were taken from the middle of a fresh toothpaste tube, while preventative gel **1** was taken from the top of a fresh tube. During the mixing step, the vial was periodically held under a stream of warm water for short time periods (a maximum time of *ca.* 10 seconds) to decrease the viscosity of the toothpaste and facilitate better mixing and dissolution of the radical. Once the radical was homogeneously mixed with the toothpaste, the sample was immediately packed into a 3.2 mm sapphire rotor. The sapphire rotor was sealed with a silicone soft plug and capped with a zirconia drive cap. All DNP samples were prepared immediately before performing NMR experiments. The maximum time it took to prepare the sample and insert the rotor into the spectrometer was *ca.* 20 – 30 minutes. We note that prolonged storage (1 – 2 weeks) of the prepared samples at *ca.* 0 °C gave no DNP enhancements due to reduction of the biradical, presumably caused by oxidation of Sn(II) to Sn(IV).

The magnetic field strength was referenced to 1 % TMS in CDCl₃ with the ¹H shift of the silicone soft plug ($\delta(^1\text{H}) = 0.24$ ppm) as a secondary chemical shift reference at 100 K. The ¹H shift of the silicone soft plug was determined based on the ¹H shift of frozen tetrachloroethane (TCE, $\delta(^1\text{H}) = 6.2$ ppm). ¹³C and ¹¹⁹Sn shifts were indirectly referenced to SiMe₄ or SnMe₄, respectively, using the previously published IUPAC recommended relative NMR frequencies.⁸⁹ All NMR spectra were initially processed and referenced with the Bruker Topspin 3.6.1 software. Carr-Purcell Meiboom-Gill (CPMG) echo trains were co-added using the NUTs NMR software (Acorn, Inc.). The ¹¹⁹Sn NMR spectra were analytically fit using the open-source ssNake NMR software.⁹¹

All experimental NMR parameters (MAS frequency, recycle delay ($\tau_{\text{rec. delay}}$), number of scans, *t*₁ dwell (Δt_1), *t*₁ TD points, *t*₁ acquisition time (*t*₁ AQ), CP/*J*-evolution durations ($\tau_{\text{CP}/J}$).

evol.) and total experimental times are given in Table S4. NMR experiments were performed with the NMR probe configured in either HXY triple-resonance mode (tuned to ^1H - ^{119}Sn - ^{13}C) or HX double-resonance mode (tuned to ^1H - ^{119}Sn). In all probe configurations, the ^1H $\pi/2$ and π pulse lengths were 2.5 and 5 μs in duration, corresponding to a 100 kHz RF field. The ^{13}C $\pi/2$ and π pulse lengths were 4 and 8 μs in duration, corresponding to a 62.5 kHz RF field. In triple-resonance HXY mode, the ^{119}Sn $\pi/2$ and π pulse lengths were 4 and 8 μs in duration, corresponding to a 62.5 kHz RF field. In double-resonance HX mode, the ^{119}Sn $\pi/2$ and π pulse lengths were 3 and 6 μs in duration, corresponding to an 83.3 kHz RF field. $^1\text{H}\rightarrow^{13}\text{C}$ cross-polarization (CP) was achieved with a 10 kHz MAS frequency with simultaneous ^1H (ramped from 90 – 100 %) and ^{13}C spin-lock pulses with RF fields of *ca.* 62 (ramped from 56 – 62 kHz) and 64 kHz, respectively. In triple-resonance HXY mode (10 kHz MAS frequency), $^1\text{H}\rightarrow^{119}\text{Sn}$ CP was achieved with simultaneous ^1H (ramped from 90 – 100 %) and ^{119}Sn spin-lock pulses with RF fields of *ca.* 72 (ramped from 65 – 72 kHz) and 56 kHz, respectively. In double-resonance HX mode (10 kHz MAS frequency), $^1\text{H}\rightarrow^{119}\text{Sn}$ CP was achieved with simultaneous ^1H (ramped from 90 – 100 %) and ^{119}Sn spin-lock pulses with RF fields of *ca.* 76 (ramped from 69 – 76 kHz) and 80 kHz, respectively. Optimization of the $^1\text{H}\rightarrow^{119}\text{Sn}$ CP contact time showed that 6 ms gave the optimal NMR signal.

All ^{119}Sn NMR spectra were acquired with CPMG detection to increase sensitivity. $\pi/2$ (1D spectra of toothpastes **1** – **4**) or π (all other spectra) pulses were implemented in the CPMG trains.⁹²⁻⁹³ 1D $^1\text{H}\rightarrow^{119}\text{Sn}$ CP-CPMG NMR spectra were acquired with multiple ^{119}Sn transmitter offsets due to the large breadth of the ^{119}Sn NMR spectra (i.e., VOCS style acquisition; Figure S11).⁸³⁻⁸⁴ 2D $^1\text{H}\rightarrow^{13}\text{C}$ and $^1\text{H}\rightarrow^{119}\text{Sn}$ CP-HETCOR NMR spectra were recorded with 100 kHz ^1H RF field of eDUMBO₁₋₂₂ homonuclear dipolar decoupling during the indirect acquisition of

^1H .⁹⁴ Each pulse in the homonuclear dipolar decoupling train was 32 μs in duration. 2D ^{119}Sn adiabatic magic-angle turning (aMAT) NMR spectra were recorded with previously described pulse sequences.⁸⁵⁻⁸⁸ Frequency swept tanh/tan inversion pulses were 100 μs in duration (i.e., 1 rotor-cycle for a 10 kHz MAS frequency) with an *ca.* 90 kHz RF field, a 2 MHz sweep width and 400 points. All ^{119}Sn aMAT spectra were recorded with $^1\text{H}\rightarrow^{119}\text{Sn}$ CP at the start of the experiment and with our previously described arbitrary indirect dwell (AID) t_1 acquisition mode to increase sensitivity.⁸⁸ 100 kHz ^1H RF field of SPINAL-64 heteronuclear decoupling was performed during the acquisition of ^{13}C and ^{119}Sn .⁹⁰

Supplementary Material

The supplementary material is available free of charge at XXXX. Additional solid-state NMR spectra, NMR analytical fitting parameters, and NMR experimental parameters. Raw 1D and 2D NMR data files in Bruker Topspin format are available at <https://doi.org/10.5281/zenodo.7569996>

Acknowledgements

This work was primarily supported by Colgate-Palmolive, Inc. Additional support for solid-state NMR experiments was provided by the National Science Foundation under Grant No. 1709972. A.J.R. acknowledges additional support from the Alfred P. Sloan Foundation through a Sloan research fellowship. Dynamic nuclear polarization solid-state NMR experiments were performed at the Ames National Laboratory. The Ames National Laboratory is operated for the U.S. DOE by Iowa State University under Contract DE-AC02-07CH11358.

Declaration of Competing Interests

C.-y.C., L.P., and Z.H. were or are currently employed by the Colgate-Palmolive Company. The Colgate-Palmolive Company provided financial support for this publication.

References

1. Satcher, D., Oral Health in America: A Report of the Surgeon General. Rockville, MD: National Institute of Dental and Craniofacial Research, National Institutes of Health, US Department of Health and Human Services, 2000.
2. Haumschild, M. S.; Haumschild, R. J., The Importance of Oral Health in Long-Term Care. *Journal of the American Medical Directors Association* **2009**, *10* (9), 667-671.
3. Frisbee, S. J.; Chambers, C. B.; Frisbee, J. C.; Goodwill, A. G.; Crout, R. J., Association Between Dental Hygiene, Cardiovascular Disease Risk Factors and Systemic Inflammation in Rural Adults. *American Dental Hygienists Association* **2010**, *84* (4), 177.
4. Satcher, D.; Nottingham, J. H., Revisiting Oral Health in America: A Report of the Surgeon General. *Am J Public Health* **2017**, *107* (S1), S32-S33.
5. Murray, J. J.; Jenkins, G. N.; Rugg-Gunn, A. J., *Fluorides in caries prevention*. Wright: Oxford, 1991.
6. Øgaard, B.; Seppä, L.; Rolla, G., Professional Topical Fluoride Applications— Clinical Efficacy and Mechanism of Action. *Advances in Dental Research* **1994**, *8* (2), 190-201.
7. Ten Cate, J. M., Current concepts on the theories of the mechanism of action of fluoride. *Acta Odontologica Scandinavica* **1999**, *57* (6), 325-329.
8. Marinho, V. C.; Higgins, J. P.; Sheiham, A.; Logan, S., Fluoride toothpastes for preventing dental caries in children and adolescents. *Cochrane Database Syst Rev* **2003**, *2003* (1), CD002278-CD002278.
9. Petersen, P. E.; Lennon, M. A., Effective use of fluorides for the prevention of dental caries in the 21st century: the WHO approach. *Community Dentistry and Oral Epidemiology* **2004**, *32* (5), 319-321.
10. Pitts, N. B.; Zero, D. T.; Marsh, P. D.; Ekstrand, K.; Weintraub, J. A.; Ramos-Gomez, F.; Tagami, J.; Twetman, S.; Tsakos, G.; Ismail, A., Dental caries. *Nature Reviews Disease Primers* **2017**, *3* (1), 17030.
11. Muhler, J. C.; Radike, A. W.; Nebergall, W. H.; Day, H. G., The Effect of a Stannous Fluoride-Containing Dentifrice on Caries Reduction in Children. *Journal of Dental Research* **1954**, *33* (5), 606-612.
12. George, A. N.; David, H. W.; Winston, D. B., Topical Applications of Sodium Fluoride and Stannous Fluoride. *Public Health Reports (1896-1970)* **1958**, *73* (9), 847-850.
13. Radike, A. W.; Gish, C. W.; Peterson, J. K.; King, J. D.; Segreto, V. A., Clinical Evaluation of Stannous Fluoride as an Anticaries Mouthrinse. *The Journal of the American Dental Association* **1973**, *86* (2), 404-408.
14. White, D. J., A 'return' to stannous fluoride dentifrices. *Journal of Clinical Dentistry* **1995**, *6* (SPEC. ISS. II), 29-36.
15. Tinanoff, N., Review of the antimicrobial action of stannous fluoride. *The Journal of clinical dentistry* **1990**, *2* (1), 22-27.
16. Ciancio, S. G., Whole mouth health. *The Journal of the American Dental Association* **2019**, *150* (4, Supplement), S1-S4.
17. Myers, C. P.; Pappas, I.; Makwana, E.; Begum-Gafur, R.; Utgikar, N.; Alsina, M. A.; Fitzgerald, M.; Trivedi, H. M.; Gaillard, J.-F.; Masters, J. G.; Sullivan, R. J., Solving the problem with stannous fluoride: Formulation, stabilization, and antimicrobial action. *The Journal of the American Dental Association* **2019**, *150* (4), S5-S13.

18. He, T.; Farrell, S., The Case for Stabilized Stannous Fluoride Dentifrice: An Advanced Formulation Designed for Patient Preference. *J. Clin. Dent.* **2017**, *28*, B1-5.
19. Lippert, F., An Introduction to Toothpaste - Its Purpose, History and Ingredients. *Monographs in Oral Science* **2013**, *23*, 1-14.
20. Hu, D.; Li, X.; Liu, H.; Mateo, L. R.; Sabharwal, A.; Xu, G.; Szewczyk, G.; Ryan, M.; Zhang, Y.-P., Evaluation of a stabilized stannous fluoride dentifrice on dental plaque and gingivitis in a randomized controlled trial with 6-month follow-up. *The Journal of the American Dental Association* **2019**, *150* (4, Supplement), S32-S37.
21. Tinanoff, N., Progress regarding the use of stannous fluoride in clinical dentistry. *The Journal of clinical dentistry* **1995**, *6 Spec No*, 37-40.
22. Mazza, J. E.; Newman, M. G.; Sims, T. N., Clinical and antimicrobial effect of stannous fluoride on periodontitis. *Journal of Clinical Periodontology* **1981**, *8* (3), 203-212.
23. Boyd, R. L.; Leggott, P. J.; Robertson, P. B., Effects on Gingivitis of Two Different 0.4% SnF₂ Gels. *Journal of Dental Research* **1988**, *67* (2), 503-507.
24. Boyd, R. L.; Chun, Y. S., Eighteen-month evaluation of the effects of a 0.4% stannous fluoride gel on gingivitis in orthodontic patients. *American Journal of Orthodontics and Dentofacial Orthopedics* **1994**, *105* (1), 35-41.
25. Weber, D. A.; Howard-Nordan, K.; Buckner, B. A.; Helsinger, S. A.; Lueders, R. A.; Court, L. K.; Bollmer, B. W.; Perlich, M. A.; Sewak, L. K., Microbiological assessment of an improved stannous fluoride dentifrice. *Journal of Clinical Dentistry* **1995**, *6* (SPEC. ISS. II), 97-104.
26. Perlich, M. A.; Bacca, L. A.; Bollmer, B. W.; Lanzalaco, A. C.; McClanahan, S. F.; Sewak, L. K.; Beiswanger, B. B.; Eichold, W. A.; Hull, J. R.; Jackson, R. D.; Mau, M. S., The clinical effect of a stabilized stannous fluoride dentifrice on plaque formation, gingivitis and gingival bleeding: A six-month study. *Journal of Clinical Dentistry* **1995**, *6* (SPEC. ISS. II), 54-58.
27. White, D. J.; Cox, E. R.; Gwynn, A. V., Effect of a stabilized stannous fluoride dentifrice on plaque acid (toxin) production. *Journal of Clinical Dentistry* **1995**, *6* (SPEC. ISS. II), 84-88.
28. Addy, M., Studies on stannous fluoride toothpaste and gel (2). Effects on salivary bacterial counts and plaque regrowth in vivo. *Journal of Clinical Periodontology* **1997**, *24* (2), 86-91.
29. Mankodi, S.; Bartizek, R. D.; Winston, J. L.; Biesbrock, A. R.; McClanahan, S. F.; He, T., Anti-gingivitis efficacy of a stabilized 0.454% stannous fluoride/sodium hexametaphosphate dentifrice: A controlled 6-month clinical trial. *Journal of Clinical Periodontology* **2005**, *32* (1), 75-80.
30. Haraszthy, V. I.; Raylae, C. C.; Sreenivasan, P. K., Antimicrobial effects of a stannous fluoride toothpaste in distinct oral microenvironments. *The Journal of the American Dental Association* **2019**, *150* (4), S14-S24.
31. Ellingsen, J. E.; Svaton, B.; Rölla, G., The effects of stannous and stannic ions on the formation and acidogenicity of dental plaque in vivo. *Acta Odontologica Scandinavica* **1980**, *38* (4), 219-222.
32. Camosci, D. A.; Tinanoff, N., Anti-bacterial Determinants of Stannous Fluoride. *Journal of Dental Research* **1984**, *63* (9), 1121-1125.
33. Rajendiran, M.; Trivedi, H. M.; Chen, D.; Gajendrareddy, P.; Chen, L., Recent Development of Active Ingredients in Mouthwashes and Toothpastes for Periodontal Diseases. *Molecules* **2021**, *26* (7).

34. Desmau, M.; Alsina, M. A.; Gaillard, J.-F., XAS study of Sn speciation in toothpaste. *Journal of Analytical Atomic Spectrometry* **2021**, *36* (2), 407-415.
35. Grey, C. P.; Dobson, C. M.; Cheetham, A. K.; Jakeman, R. J. B., Studies of rare-earth stannates by tin-119 MAS NMR. The use of paramagnetic shift probes in the solid state. *Journal of the American Chemical Society* **1989**, *111* (2), 505-511.
36. Clayden, N. J.; Dobson, C. M.; Fern, A., High-resolution solid-state tin-119 nuclear magnetic resonance spectroscopy of ternary tin oxides. *Journal of the Chemical Society, Dalton Transactions* **1989**, (5), 843-847.
37. Mitchell, T. N., Solid state NMR spectroscopy of tin compounds. In *Chemistry of Tin*, Smith, P. J., Ed. Springer Netherlands: Dordrecht, 1998; pp 480-495.
38. Wrackmeyer, B., Application of ^{119}Sn NMR Parameters. In *Annual Reports on NMR Spectroscopy*, Webb, G. A., Ed. Academic Press: 1999; Vol. 38, pp 203-264.
39. Eichler, B. E.; Phillips, B. L.; Power, P. P.; Augustine, M. P., Solid-State and High-Resolution Liquid ^{119}Sn NMR Spectroscopy of Some Monomeric, Two-Coordinate Low-Valent Tin Compounds: Very Large Chemical Shift Anisotropies. *Inorganic Chemistry* **2000**, *39* (24), 5450-5453.
40. MacKenzie, K. J. D.; Smith, M. E., NMR of Other Spin- $\frac{1}{2}$ Nuclei. In *Multinuclear Solid-State NMR of Inorganic Materials*, Pergamon: Pergamon Materials Series, 2002; Vol. 6, pp 535-625.
41. Amornsakchai, P.; Apperley, D. C.; Harris, R. K.; Hodgkinson, P.; Waterfield, P. C., Solid-state NMR studies of some tin(II) compounds. *Solid State Nuclear Magnetic Resonance* **2004**, *26* (3), 160-171.
42. Agustin, D.; Ehses, M., ^{119}Sn NMR spectroscopic and structural properties of transition metal complexes with terminal stannylene ligands. *Comptes Rendus Chimie* **2009**, *12* (10), 1189-1227.
43. Wrackmeyer, B., Germanium, Tin, and Lead NMR. In *eMagRes*, Harris, R. K.; Wasylishen, R. L., Eds. 2011, <https://doi.org/10.1002/9780470034590.emrstm0190.pub2>.
44. Mitchell, M. R.; Reader, S. W.; Johnston, K. E.; Pickard, C. J.; Whittle, K. R.; Ashbrook, S. E., ^{119}Sn MAS NMR and first-principles calculations for the investigation of disorder in stannate pyrochlores. *Physical Chemistry Chemical Physics* **2011**, *13* (2), 488-497.
45. Frerichs, J. E.; Koppe, J.; Engelbert, S.; Heletta, L.; Brunklaus, G.; Winter, M.; Madsen, G. K. H.; Hansen, M. R., ^{119}Sn and ^7Li Solid-State NMR of the Binary Li-Sn Intermetallics: Structural Fingerprinting and Impact on the Isotropic ^{119}Sn Shift via DFT Calculations. *Chemistry of Materials* **2021**, *33* (10), 3499-3514.
46. Mason, J., Conventions for the reporting of nuclear magnetic shielding (or shift) tensors suggested by participants in the NATO ARW on NMR shielding constants at the University of Maryland, College Park, July 1992. *Solid State Nucl. Magn. Reson.* **1993**, *2* (5), 285-288.
47. Jameson, C. J., Reply to 'conventions for tensor quantities used in nuclear magnetic resonance, nuclear quadrupole resonance and electron spin resonance spectroscopy. *Solid State Nucl. Magn. Reson.* **1998**, *11* (3), 265-268.
48. Cossement, C.; Darville, J.; Gilles, J.-M.; Nagy, J. B.; Fernandez, C.; Amoureux, J.-P., Chemical shift anisotropy and indirect coupling in SnO_2 and SnO . *Magnetic Resonance in Chemistry* **1992**, *30* (3), 263-270.
49. Mundus, C.; Taillades, G.; Pradel, A.; Ribes, M., A ^{119}Sn solid-state nuclear magnetic resonance study of crystalline tin sulphides. *Solid State Nuclear Magnetic Resonance* **1996**, *7* (2), 141-146.

50. Gay, I. D.; Jones, C. H. W.; Sharma, R. D., A multinuclear solid-state NMR study of the dimethyltin chalcogenides $((\text{CH}_3)_2\text{SnE})_3$, E = S, Se, Te. *J. Magn. Reson.* **1989**, *84* (3), 501-514.
51. Lyčka, A.; Holeček, J.; Schneider, B.; Straka, J., High-resolution solid-state ^{119}Sn NMR spectroscopy of some organotin(IV) oxinates and thiooxinates. *Journal of Organometallic Chemistry* **1990**, *389* (1), 29-39.
52. Bernatowicz, P.; Dinnebier, R. E.; Helluy, X.; Kümmerlen, J.; Sebald, A., Dynamic disorder in solid tetrakis(trimethylstannyl)methane, $\text{C}(\text{SnMe}_3)_4$, investigated by one- and two-dimensional variable-temperature ^{119}Sn and ^{13}C NMR spectroscopy. *Applied Magnetic Resonance* **1999**, *17* (2), 385-398.
53. Knyrim, J. S.; Schappacher, F. M.; Pöttgen, R.; Schmedt auf der Günne, J.; Johrendt, D.; Huppertz, H., Pressure-Induced Crystallization and Characterization of the Tin Borate $\beta\text{-SnB}_4\text{O}_7$. *Chemistry of Materials* **2007**, *19* (2), 254-262.
54. Catalano, J.; Murphy, A.; Yao, Y.; Alkan, F.; Zumbulyadis, N.; Centeno, S. A.; Dybowski, C., ^{207}Pb and ^{119}Sn Solid-State NMR and Relativistic Density Functional Theory Studies of the Historic Pigment Lead–Tin Yellow Type I and Its Reactivity in Oil Paintings. *The Journal of Physical Chemistry A* **2014**, *118* (36), 7952-7958.
55. Wolf, P.; Valla, M.; Rossini, A. J.; Comas-Vives, A.; Núñez-Zarur, F.; Malaman, B.; Lesage, A.; Emsley, L.; Copéret, C.; Hermans, I., NMR Signatures of the Active Sites in $\text{Sn-}\beta$ Zeolite. *Angewandte Chemie International Edition* **2014**, *53* (38), 10179-10183.
56. Zilm, K. W.; Lawless, G. A.; Merrill, R. M.; Millar, J. M.; Webb, G. G., Nature of the tin-tin double bond as studied by solid-state and solution nuclear magnetic resonance. *Journal of the American Chemical Society* **1987**, *109* (23), 7236-7238.
57. Chaudhuri, S.; Wang, F.; Grey, C. P., Resolving the Different Dynamics of the Fluorine Sublattices in the Anionic Conductor BaSnF_4 by Using High-Resolution MAS NMR Techniques. *Journal of the American Chemical Society* **2002**, *124* (39), 11746-11757.
58. Spikes, G. H.; Giuliani, J. R.; Augustine, M. P.; Nowik, I.; Herber, R. H.; Power, P. P., Solid-State ^{119}Sn NMR and Mössbauer Spectroscopy of “Distannynes”: Evidence for Large Structural Differences in the Crystalline Phase. *Inorganic Chemistry* **2006**, *45* (22), 9132-9136.
59. Bräuniger, T.; Ghedia, S.; Jansen, M., Covalent Bonds in $\alpha\text{-SnF}_2$ Monitored by J-Couplings in Solid-State NMR Spectra. *Zeitschrift für anorganische und allgemeine Chemie* **2010**, *636* (13-14), 2399-2404.
60. Krebs, K. M.; Wiederkehr, J.; Schneider, J.; Schubert, H.; Eichele, K.; Wesemann, L., η^3 -Allyl Coordination at Tin(II)—Reactivity towards Alkynes and Benzonitrile. *Angewandte Chemie International Edition* **2015**, *54* (18), 5502-5506.
61. Pöppler, A.-C.; Demers, J.-P.; Malon, M.; Singh, A. P.; Roesky, H. W.; Nishiyama, Y.; Lange, A., Ultrafast Magic-Angle Spinning: Benefits for the Acquisition of Ultrawide-Line NMR Spectra of Heavy Spin- Nuclei. *ChemPhysChem* **2016**, *17* (6), 812-816.
62. Ha, M.; Karmakar, A.; Bernard, G. M.; Basilio, E.; Krishnamurthy, A.; Askar, A. M.; Shankar, K.; Kroeker, S.; Michaelis, V. K., Phase Evolution in Methylammonium Tin Halide Perovskites with Variable Temperature Solid-State ^{119}Sn NMR Spectroscopy. *The Journal of Physical Chemistry C* **2020**, *124* (28), 15015-15027.
63. Wrackmeyer, B.; Kehr, G.; Sebald, A.; Kümmerlen, J., Organotin cations stabilized by π coordination – synthesis and NMR studies in solution and in the solid state. *Chemische Berichte* **1992**, *125* (7), 1597-1603.
64. Macdonald, C. L. B.; Bandyopadhyay, R.; Cooper, B. F. T.; Friedl, W. W.; Rossini, A. J.; Schurko, R. W.; Eichhorn, S. H.; Herber, R. H., Experimental and Computational Insights into

the Stabilization of Low-Valent Main Group Elements Using Crown Ethers and Related Ligands. *Journal of the American Chemical Society* **2012**, *134* (9), 4332-4345.

65. Kubicki, D. J.; Prochowicz, D.; Salager, E.; Rakhmatullin, A.; Grey, C. P.; Emsley, L.; Stranks, S. D., Local Structure and Dynamics in Methylammonium, Formamidinium, and Cesium Tin(II) Mixed-Halide Perovskites from ^{119}Sn Solid-State NMR. *Journal of the American Chemical Society* **2020**, *142* (17), 7813-7826.
66. Maly, T.; Debelouchina, G. T.; Bajaj, V. S.; Hu, K.-N.; Joo, C.-G.; Mak-Jurkauskas, M. L.; Sirigiri, J. R.; van der Wel, P. C. A.; Herzfeld, J.; Temkin, R. J.; Griffin, R. G., Dynamic nuclear polarization at high magnetic fields. *J. Chem. Phys.* **2008**, *128* (5), 052211.
67. Rosay, M.; Tometich, L.; Pawsey, S.; Bader, R.; Schauwecker, R.; Blank, M.; Borchard, P. M.; Cauffman, S. R.; Felch, K. L.; Weber, R. T.; Temkin, R. J.; Griffin, R. G.; Maas, W. E., Solid-state dynamic nuclear polarization at 263 GHz: spectrometer design and experimental results. *Phys. Chem. Chem. Phys.* **2010**, *12* (22), 5850-5860.
68. Ni, Q. Z.; Daviso, E.; Can, T. V.; Markhasin, E.; Jawla, S. K.; Swager, T. M.; Temkin, R. J.; Herzfeld, J.; Griffin, R. G., High Frequency Dynamic Nuclear Polarization. *Acc. Chem. Res.* **2013**, *46* (9), 1933-1941.
69. Su, Y.; Andreas, L.; Griffin, R. G., Magic Angle Spinning NMR of Proteins: High-Frequency Dynamic Nuclear Polarization and ^1H Detection. *Annu. Rev. Biochem.* **2015**, *84* (1), 465-497.
70. Conley, M. P.; Rossini, A. J.; Comas-Vives, A.; Valla, M.; Casano, G.; Ouari, O.; Tordo, P.; Lesage, A.; Emsley, L.; Copéret, C., Silica-surface reorganization during organotin grafting evidenced by ^{119}Sn DNP SENS: a tandem reaction of gem-silanols and strained siloxane bridges. *Physical Chemistry Chemical Physics* **2014**, *16* (33), 17822-17827.
71. Protesescu, L.; Rossini, A. J.; Kriegner, D.; Valla, M.; de Kergommeaux, A.; Walter, M.; Kravchyk, K. V.; Nachttegaal, M.; Stangl, J.; Malaman, B.; Reiss, P.; Lesage, A.; Emsley, L.; Copéret, C.; Kovalenko, M. V., Unraveling the Core-Shell Structure of Ligand-Capped Sn/SnOx Nanoparticles by Surface-Enhanced Nuclear Magnetic Resonance, Mössbauer, and X-ray Absorption Spectroscopies. *ACS Nano* **2014**, *8* (3), 2639-2648.
72. Gunther, W. R.; Michaelis, V. K.; Caporini, M. A.; Griffin, R. G.; Román-Leshkov, Y., Dynamic Nuclear Polarization NMR Enables the Analysis of Sn-Beta Zeolite Prepared with Natural Abundance ^{119}Sn Precursors. *Journal of the American Chemical Society* **2014**, *136* (17), 6219-6222.
73. Wolf, P.; Liao, W.-C.; Ong, T.-C.; Valla, M.; Harris, J. W.; Gounder, R.; van der Graaff, W. N. P.; Pidko, E. A.; Hensen, E. J. M.; Ferrini, P.; Dijkmans, J.; Sels, B.; Hermans, I.; Copéret, C., Identifying Sn Site Heterogeneities Prevalent Among Sn-Beta Zeolites. *Helvetica Chimica Acta* **2016**, *99* (12), 916-927.
74. Harris, J. W.; Liao, W.-C.; Di Iorio, J. R.; Henry, A. M.; Ong, T.-C.; Comas-Vives, A.; Copéret, C.; Gounder, R., Molecular Structure and Confining Environment of Sn Sites in Single-Site Chabazite Zeolites. *Chemistry of Materials* **2017**, *29* (20), 8824-8837.
75. Wolf, P.; Valla, M.; Núñez-Zarur, F.; Comas-Vives, A.; Rossini, A. J.; Firth, C.; Kallas, H.; Lesage, A.; Emsley, L.; Copéret, C.; Hermans, I., Correlating Synthetic Methods, Morphology, Atomic-Level Structure, and Catalytic Activity of Sn- β Catalysts. *ACS Catalysis* **2016**, *6* (7), 4047-4063.
76. Sauvée, C.; Rosay, M.; Casano, G.; Aussenac, F.; Weber, R. T.; Ouari, O.; Tordo, P., Highly Efficient, Water-Soluble Polarizing Agents for Dynamic Nuclear Polarization at High Frequency. *Angewandte Chemie International Edition* **2013**, *52* (41), 10858-10861.

77. Bork, M.; Hoppe, R., Zum Aufbau von PbF₄ mit Strukturverfeinerung an SnF₄. *Zeitschrift für anorganische und allgemeine Chemie* **1996**, 622 (9), 1557-1563.
78. McDonald, R. C.; Hau, H. H. K.; Eriks, K., Crystallographic studies of tin(II) compounds. I. Crystal structure of tin(II) fluoride, SnF₂. *Inorganic Chemistry* **1976**, 15 (4), 762-765.
79. Neue, G.; Bai, S.; Taylor, R. E.; Beckmann, P. A.; Vega, A. J.; Dybowski, C., ¹¹⁹Sn spin-lattice relaxation in α -SnF₂. *Physical Review B* **2009**, 79 (21), 214302.
80. Gilbert, T. M.; Ziegler, T., Prediction of ¹⁹⁵Pt NMR Chemical Shifts by Density Functional Theory Computations: The Importance of Magnetic Coupling and Relativistic Effects in Explaining Trends. *The Journal of Physical Chemistry A* **1999**, 103 (37), 7535-7543.
81. Jokisaari, J.; Järvinen, S.; Autschbach, J.; Ziegler, T., ¹⁹⁹Hg Shielding Tensor in Methylmercury Halides: NMR Experiments and ZORA DFT Calculations. *The Journal of Physical Chemistry A* **2002**, 106 (40), 9313-9318.
82. Werbeck, N. D.; Hansen, D. F., Heteronuclear transverse and longitudinal relaxation in AX₄ spin systems: Application to ¹⁵N relaxations in ¹⁵NH₄⁺. *Journal of Magnetic Resonance* **2014**, 246, 136-148.
83. Massiot, D.; Farnan, I.; Gautier, N.; Trumeau, D.; Trokiner, A.; Coutures, J. P., ⁷¹Ga and ⁶⁹Ga nuclear magnetic resonance study of β -Ga₂O₃: resolution of four- and six-fold coordinated Ga sites in static conditions. *Solid State Nuclear Magnetic Resonance* **1995**, 4 (4), 241-248.
84. Schurko, R. W.; Wi, S.; Frydman, L., Dynamic Effects on the Powder Line Shapes of Half-Integer Quadrupolar Nuclei: A Solid-State NMR Study of XO₄⁻ Groups. *The Journal of Physical Chemistry A* **2002**, 106 (1), 51-62.
85. Hu, J. Z.; Wang, W.; Liu, F.; Solum, M. S.; Alderman, D. W.; Pugmire, R. J.; Grant, D. M., Magic-Angle-Turning Experiments for Measuring Chemical-Shift-Tensor Principal Values in Powdered Solids. *Journal of Magnetic Resonance, Series A* **1995**, 113 (2), 210-222.
86. Clément, R. J.; Pell, A. J.; Middlemiss, D. S.; Strobridge, F. C.; Miller, J. K.; Whittingham, M. S.; Emsley, L.; Grey, C. P.; Pintacuda, G., Spin-Transfer Pathways in Paramagnetic Lithium Transition-Metal Phosphates from Combined Broadband Isotropic Solid-State MAS NMR Spectroscopy and DFT Calculations. *Journal of the American Chemical Society* **2012**, 134 (41), 17178-17185.
87. Perras, F. A.; Venkatesh, A.; Hanrahan, M. P.; Goh, T. W.; Huang, W.; Rossini, A. J.; Pruski, M., Indirect detection of infinite-speed MAS solid-state NMR spectra. *Journal of Magnetic Resonance* **2017**, 276, 95-102.
88. Venkatesh, A.; Perras, F. A.; Rossini, A. J., Proton-Detected Solid-State NMR Spectroscopy of Spin-1/2 Nuclei with Large Chemical Shift Anisotropy. *J. Magn. Reson.* **2021**, 327, 106983.
89. Harris, R. K.; Becker, E. D.; Cabral de Menezes, S. M.; Goodfellow, R.; Granger, P., NMR Nomenclature: Nuclear Spin Properties and Conventions for Chemical Shifts: IUPAC Recommendations 2001. *Solid State Nucl. Magn. Reson.* **2002**, 22 (4), 458-483.
90. Fung, B. M.; Khitrin, A. K.; Ermolaev, K., An Improved Broadband Decoupling Sequence for Liquid Crystals and Solids. *J. Magn. Reson.* **2000**, 142 (1), 97-101.
91. van Meerten, S. G. J.; Franssen, W. M. J.; Kentgens, A. P. M., ssNake: A Cross-Platform Open-Source NMR Data Processing and Fitting Application. *J. Magn. Reson.* **2019**, 301, 56-66.
92. Siegel, R.; Nakashima, T. T.; Wasylishen, R. E., Application of Multiple-Pulse Experiments to Characterize Broad NMR Chemical-Shift Powder Patterns from Spin-1/2 Nuclei in the Solid State. *The Journal of Physical Chemistry B* **2004**, 108 (7), 2218-2226.

93. Altenhof, A. R.; Jaroszewicz, M. J.; Lindquist, A. W.; Foster, L. D. D.; Veinberg, S. L.; Schurko, R. W., Practical Aspects of Recording Ultra-Wideline NMR Patterns under Magic-Angle Spinning Conditions. *The Journal of Physical Chemistry C* **2020**, *124* (27), 14730-14744.
94. Sakellariou, D.; Lesage, A.; Hodgkinson, P.; Emsley, L., Homonuclear dipolar decoupling in solid-state NMR using continuous phase modulation. *Chem. Phys. Lett.* **2000**, *319* (3), 253-260.

Supplementary Material

Structural Characterization of Tin in Toothpaste By Dynamic Nuclear Polarization Enhanced ^{119}Sn Solid-State NMR Spectroscopy

Rick W. Dorn,^{1,2} Scott L. Carnahan,^{1,2} Chi-yuan Chen,³ Long Pan,³ Zhigang Hao,^{3} Aaron J.
Rossini,^{1,2*}*

¹US Department of Energy, Ames National Laboratory, Ames, IA, USA, 50011.

²Iowa State University, Department of Chemistry, Ames, IA, USA, 50011.

³Colgate-Palmolive Company, Piscataway, NJ, USA 08855.

AUTHOR INFORMATION

Corresponding Author

*e-mail: zhigang_hao@colpal.com, phone: 732-878-6218.

*e-mail: arossini@iastate.edu, phone: 515-294-8952.

Table of Contents

	Page
Supplementary Figures	
Figure S1. 2D $^{19}\text{F}\{^{119}\text{Sn}\}$ J -HMQC spectrum of SnF_4	S3
Figure S2. 1D ^{19}F NMR spectra of SnF_2 and SnF_4	S4
Figure S3. 2D $^{19}\text{F}\{^{119}\text{Sn}\}$ J -HMQC spectrum of SnF_2	S5
Figure S4. 1D ^{19}F solution NMR spectra of the model toothpaste	S6
Figure S5. 2D solution $^{19}\text{F}\{^{119}\text{Sn}\}$ J -HMQC spectrum of the model toothpaste	S7
Figure S6. 1D room temperature ^{119}Sn NMR spectra	S8
Figure S7. $^1\text{H}\rightarrow^{13}\text{C}$ CPMAS DNP enhancements	S10
Figure S8. $^1\text{H}\rightarrow^{119}\text{Sn}$ CP-CPMG DNP enhancements	S11
Figure S9. $^1\text{H}\rightarrow^{13}\text{C}$ and $^1\text{H}\rightarrow^{119}\text{Sn}$ CP-HETCOR spectra of preventative gel 1	S12
Figure S10. Mass spectra of SnF_2 -glycerol complex	S13
Figure S11. $^1\text{H}\rightarrow^{119}\text{Sn}$ CP-CPMG VOCS spectra of toothpaste samples	S14
Figure S12. 2D ^{119}Sn aMAT spectrum of preventative gel 1	S15
Figure S13. 2D ^{119}Sn aMAT spectrum of toothpaste 2	S16
Figure S14. 2D ^{119}Sn aMAT spectrum of toothpaste 3	S17
Figure S15. 2D ^{119}Sn aMAT spectrum of toothpaste 4	S18
Figure S16. Effect of the number of echo trains during the acquisition of $^1\text{H}\rightarrow^{119}\text{Sn}$ CP-CPMG spectra	S20
Figure S17. ^{119}Sn MAS NMR spectra of SnF_2 obtained with different MAS frequencies.	S21
Supplementary Tables	
Table S1. DFT Calculated ^{119}Sn and ^{19}F NMR Parameters of SnF_2 and SnF_4	S2
Table S2. Major toothpaste ingredients	S9
Table S3. ^{119}Sn CP-CPMG fitting parameters	S19
Table S4. Experimental NMR parameters	S22-S23

Table S1. Periodic plane-wave GIPAW DFT calculated ^{119}Sn and ^{19}F NMR parameters of SnF_2 and SnF_4 .

Species	Site/Nucleus	σ_{iso} (ppm) ^a	Ω (ppm)	κ
^{119}Sn				
SnF_4	SnF_6	3478.4	545.8	-1.0
SnF_2	SnF_3	3517.5	661.5	-0.37
	SnF_5	3594.3	593.4	-0.41
^{19}F				
SnF_4	1	297	-	-
	2	271	-	-
SnF_2	1	168	-	-
	2	169	-	-
	3	180	-	-
	4	191	-	-

^aRelative differences in isotropic shielding (σ_{iso}) are theoretically the same as relative differences in isotropic shift (δ_{iso}). A higher shielding value means that the chemical shift will be lower.

Figure S1. (Left) 2D $^{19}\text{F}\{^{119}\text{Sn}\}$ J-HMQC solid-state NMR spectrum of SnF_4 recorded with 320 μs of total J -evolution and a 25 kHz MAS frequency. (Right) ^{119}Sn NMR spectra extracted from the rows of 2D $^{19}\text{F}\{^{119}\text{Sn}\}$ J-HMQC spectrum at the indicated ^{119}Sn chemical shifts.

Figure S2. Comparison of 1D ^{19}F solid-state NMR spectra of (upper to lower) SnF_4 , SnF_2 from supplier **a**, and SnF_2 from supplier **b** (directly from the bottle or ball-milled). Asterisks (*) correspond to spinning sidebands. Spectra were recorded with an MAS frequency of 25 kHz.

Figure S3. 2D $^{19}\text{F}\{^{119}\text{Sn}\}$ J -HMQC solid-state NMR spectrum of ball-milled SnF_2 from supplier **b** recorded with a 25 kHz MAS frequency and 80 μs of total J -evolution. Asterisks (*) indicate spinning sidebands. The reduction in signal intensity for the SnF_5 site presumably arises from decreased values of $^1J(^{119}\text{Sn}-^{19}\text{F})$ and/or reduced ^{19}F transverse relaxation times.

Figure S4. 1D ^{19}F spin echo solution NMR spectra the model toothpaste recorded with a (upper) 2 ms or (lower) 0.1 ms echo period. The broad hump from *ca.* -150 to -200 ppm are background ^{19}F NMR signals from Teflon within the probe.

Figure S5. 2D $^{19}\text{F}\{^{119}\text{Sn}\}$ J -HMQC solution NMR spectrum of the model toothpaste. Both sets of ^{119}Sn NMR signals exhibit multiple peaks because of truncation of the indirect dimension NMR signals.

Figure S6. Comparison of (upper two) ^{119}Sn spin echo and $^{19}\text{F}\{^{119}\text{Sn}\}$ J -HMQC solution NMR spectra of the model toothpaste with that of (lower two) ^{119}Sn spin echo solid-state NMR spectra of SnF_4 and SnF_2 (supplier **b**). Asterisks indicate spinning sidebands.

Table S2. Major ingredients in preventative gel **1** and toothpastes **1-4**.

Sample	Main Solvent	SnF ₂ Stabilizer
Preventative gel 1	Glycerin	No
Toothpaste 1	Glycerin	No
Toothpaste 2	Water, Sorbitol, Glycerin	No
Toothpaste 3	Glycerin	No
Toothpaste 4	Water, Sorbitol	SnCl ₂

Figure S7. $^1\text{H} \rightarrow ^{13}\text{C}$ CPMAS spectra of (A) the model toothpaste and (B) preventative gel 1 recorded (black) with or (red) without microwave (MW) irradiation of the electron spins. The DNP enhancements (ϵ) are given in the figure. The intensity of the spectra without microwave irradiation was increased by a factor 64.

Figure S8. $^1\text{H} \rightarrow ^{119}\text{Sn}$ CP-CPMG NMR spectra of the (A) model toothpaste and toothpastes (B) 1, (C) 2, (D) 3 and (E) 4 recorded (black) with or (red) without microwave (MW) irradiation of the electron spins. The estimated DNP enhancements (ϵ) are given in the figure.

Figure S9. DNP-enhanced 2D (A) $^1\text{H} \rightarrow ^{13}\text{C}$ and (B) $^1\text{H} \rightarrow ^{119}\text{Sn}$ CP-HETCOR NMR spectra of preventative gel 1 record with a 10 kHz MAS frequency, eDUMBO₁₋₂₂ ^1H homonuclear decoupling during the indirect acquisition of ^1H , and (B) CPMG detection of ^{119}Sn .

Figure S10. Mass spectra of a Sn^{2+} -glycerol complex ($C_3H_5O_3Sn_1$). When a solution of 0.45 wt.% SnF_2 in glycerol was made and heated to 60 °C, the Sn^{2+} -glycerol complex could be detected with direct injection mode on a liquid chromatography-high resolution mass spectrometer. The delivery solvent was 50% methanol-water.

Figure S11. DNP-enhanced $^1\text{H} \rightarrow ^{119}\text{Sn}$ CP-CPMG NMR spectra of (upper to lower) the model toothpaste, preventative gel **1**, and toothpastes **1**, **2**, **3** and **4** recorded with a 10 kHz MAS frequency and variable ^{119}Sn transmitter offsets (i.e., VOCS style acquisition). (Colored) Sub-spectra for specific ^{119}Sn transmitter offsets are shown below the (black, upper) summed spectra.

Figure S12. (A) DNP-enhanced 2D ^{119}Sn aMAT NMR spectrum of preventative gel 1 acquired with a 10 kHz MAS frequency, $^1\text{H} \rightarrow ^{119}\text{Sn}$ CP at the start of the experiment, and CPMG for ^{119}Sn detection. (B) ^{119}Sn NMR spectra extracted from the 2D aMAT NMR spectrum at the indicated ^{119}Sn isotropic chemical shifts (δ_{iso}). Analytically simulated spectra are shown (colored) below the (black) experimental MAS spectra.

Figure S13. (A) DNP-enhanced 2D ^{119}Sn aMAT NMR spectrum of toothpaste **2** acquired with a 10 kHz MAS frequency, $^1\text{H} \rightarrow ^{119}\text{Sn}$ CP at the start of the experiment, and CPMG for ^{119}Sn detection. (B) ^{119}Sn NMR spectra extracted from the 2D aMAT NMR spectrum at the indicated ^{119}Sn isotropic chemical shifts (δ_{iso}). Analytically simulated spectra are shown (colored) below the (black) experimental MAS spectra.

Figure S14. (A) DNP-enhanced 2D ^{119}Sn aMAT NMR spectrum of toothpaste **3** acquired with a 10 kHz MAS frequency, $^1\text{H} \rightarrow ^{119}\text{Sn}$ CP at the start of the experiment, and CPMG for ^{119}Sn detection. (B) ^{119}Sn NMR spectra extracted from the 2D aMAT NMR spectrum at the indicated ^{119}Sn isotropic chemical shifts (δ_{iso}). Analytically simulated spectra are shown (colored) below the (black) experimental MAS spectra.

Figure S15. (A) DNP-enhanced 2D ^{119}Sn aMAT NMR spectrum of toothpaste **4** acquired with a 10 kHz MAS frequency, $^1\text{H} \rightarrow ^{119}\text{Sn}$ CP at the start of the experiment, and CPMG for ^{119}Sn detection. (B) ^{119}Sn NMR spectra extracted from the 2D aMAT NMR spectrum at the indicated ^{119}Sn isotropic chemical shifts (δ_{iso}). Analytically simulated spectra are shown (colored) below the (black) experimental MAS spectra.

Table S3. ¹¹⁹Sn CP-CPMG NMR Spectral Fitting Parameters of Toothpaste Samples

Sn Oxidation State	δ_{iso} (ppm)	δ_{iso} distribution (ppm) ^b	Ω (ppm)	κ	Population (%)
Model Toothpaste					
+4	-675	134	150	0 ^c	8
+2	-800	134	1200	0.7	50
+2	-650	134	1200	0.7	42
Preventative Gel 1					
+4	-800	67	150	0	3
+4	-720	67	150	0	4
+2	-820	134	1200	0.7	67
+2	-650	134	1200	0.7	26
Toothpaste 1					
+4	-750	67	150	0 ^c	12
+4	-655	67	250	0 ^c	8
+2	-630	201	1200	0.7	80
Toothpaste 2 (Fresh)					
+4	-738	54	150	0 ^c	7
+4	-655	50	250	0 ^c	3
+2	-670	201	1150	0.7	90
Toothpaste 2 (1 Day Air Exposed)					
+4	-738	54	150	0 ^c	10
+4	-655	50	250	0 ^c	7
+2	-670	201	1150	0.7	83
Toothpaste 3					
+4	-749	67	150	0 ^c	13
+4	-655	54	250	0 ^c	7
+2	-630	201	1200	0.7	80
Toothpaste 4					
+4	-639	33	220	0 ^c	6
+4	-550	33	220	0 ^c	5
+4	-475	33	220	0 ^c	21
+2	-475	134	1100	0.7	68

^aColor of text for each site corresponds to the color of the fit in Figure 5 of the main text.

^bDistribution in δ_{iso} was determined from the amount of Gaussian line-broadening used in the fit, where the distribution is equal to $\pm 1\sigma$ (i.e., 68.2 % of the area in the Gaussian curve). ^cCSA is too small to accurately determine κ .

Figure S16. DNP-enhanced $^1\text{H} \rightarrow ^{119}\text{Sn}$ CP-CPMG NMR spectra of (A) the model toothpaste and (B) toothpaste **1**. The number of echo trains used during CPMG acquisition is indicated within the figure. Different number of echo trains were obtained by manipulating the FID recorded with either (A) 50 or (B) 100 echoes.

Figure S17. ¹¹⁹Sn solid-state NMR spectra of SnF₂ obtained with MAS frequencies of 25 kHz (upper) and 20 kHz (lower) to assign isotropic signals and spinning sidebands. Analytical simulations are shown below each spectrum and the best fit CS tensor parameters are indicated.

Table S4. Experimental NMR parameters.

Figure	Expt.	MAS (kHz)	$\tau_{\text{rec. delay}}$ (s)	# of Scans	Δt_1 (μs)	t_1 TD Points	t_1 AQ (ms)	$\tau_{\text{CP}/\text{evol.}}$ (ms)	Total Expt (h)
1B (upper)	^{119}Sn Spin Echo	25	150	400	-	-	-	-	16.7
1B (middle)	^{119}Sn Spin Echo	25	50	848	-	-	-	-	11.8
1B (lower)	^{119}Sn Spin Echo	25	100	384	-	-	-	-	10.7
1C (upper)	2D $^{19}\text{F}\{^{119}\text{Sn}\}$ J -HMQC	25	5	32	2.5	256	0.32	0.32 ^a	11.4
1C (lower)	2D $^{19}\text{F}\{^{119}\text{Sn}\}$ J -HMQC	20	10	64	4.0	128	0.256	0.1 ^a	22.8
2A (upper)	^{19}F Spin Echo	Solution	20 ^b	8	-	-	-	-	0.04
2A (middle)	^{19}F Spin Echo	25	30 ^b	16	-	-	-	-	0.13
2A (lower)	^{19}F Spin Echo	25	750 ^c	4	-	-	-	-	0.83
2C	2D $^{19}\text{F}\{^{119}\text{Sn}\}$ J -HSQC	Solution	0.4	8	20	320	3.2	0.28 ^a	0.28
3A (MW On)	$^1\text{H} \rightarrow ^{13}\text{C}$ CPMAS	10	20	4	-	-	-	1 ^d	0.02
3A (MW Off)	$^1\text{H} \rightarrow ^{13}\text{C}$ CPMAS	10	20	32	-	-	-	1 ^d	0.18
3B (MW On)	$^1\text{H} \rightarrow ^{119}\text{Sn}$ CP-CPMG	10	20	4	-	-	-	6 ^d	0.02
3B (MW Off)	$^1\text{H} \rightarrow ^{119}\text{Sn}$ CP-CPMG	10	20	128	-	-	-	6 ^d	0.71
3C	2D $^1\text{H} \rightarrow ^{13}\text{C}$ CP-HETCOR	10	2	4	64	256	8.192	1 ^d	0.57
3D	2D $^1\text{H} \rightarrow ^{119}\text{Sn}$ CP-HETCOR	10	20	16	64	128	4.096	6 ^d	11.4
4A	2D ^{119}Sn aMAT	10	7.28	14	3.4	200	0.34	2 ^d	5.7
4C	2D ^{119}Sn aMAT	10	4	112	3.4	124	0.211	6 ^d	15.4
5 (model)	$^1\text{H} \rightarrow ^{119}\text{Sn}$ CP-CPMG	10	20	64 (5) ^e	-	-	-	6 ^d	1.8 ^f
5 (Gel 1)	$^1\text{H} \rightarrow ^{119}\text{Sn}$ CP-CPMG	10	15.9	256 (5) ^e	-	-	-	6 ^d	5.6 ^f
5 (Toothpaste 1)	$^1\text{H} \rightarrow ^{119}\text{Sn}$ CP-CPMG	10	4	256 (4) ^e	-	-	-	6 ^d	1.1 ^f
5 (Toothpaste 2)	$^1\text{H} \rightarrow ^{119}\text{Sn}$ CP-CPMG	10	4	512 (4) ^e	-	-	-	6 ^d	2.3 ^f
5 (Toothpaste 2 - air)	$^1\text{H} \rightarrow ^{119}\text{Sn}$ CP-CPMG	10	8	384 (4) ^e	-	-	-	6 ^d	3.4 ^f
5 (Toothpaste 3)	$^1\text{H} \rightarrow ^{119}\text{Sn}$ CP-CPMG	10	4	256 (4) ^e	-	-	-	6 ^d	1.1 ^f
5 (Toothpaste 4)	$^1\text{H} \rightarrow ^{119}\text{Sn}$ CP-CPMG	10	2	1024 (3) ^e	-	-	-	6 ^d	1.7 ^f
S1	2D $^{19}\text{F}\{^{119}\text{Sn}\}$ J -HMQC	25	2	64	10	128	0.64	0.32 ^a	4.6
S2 (Supplier a)	^{19}F Spin Echo	25	50 ^b	16	-	-	-	-	0.22
S2 (Supplier b ball milled)	^{19}F Spin Echo	25	100 ^b	4	-	-	-	-	0.11
S3	2D $^{19}\text{F}\{^{119}\text{Sn}\}$ J -HMQC	25	10	32	4	128	0.256	0.08 ^a	11.4
S5	2D $^{19}\text{F}\{^{119}\text{Sn}\}$ J -HMQC	Solution	0.5	8	20	160	1.6	1.2 ^a	0.18
S6 (upper)	^{119}Sn Spin Echo	Solution	1	128	-	-	-	-	0.04
S7B	$^1\text{H} \rightarrow ^{13}\text{C}$ CPMAS	10	5	4	-	-	-	1 ^d	0.01
S8B (MW on)	$^1\text{H} \rightarrow ^{119}\text{Sn}$ CP-CPMG	10	4	256	-	-	-	6 ^d	0.28
S8B (MW off)	$^1\text{H} \rightarrow ^{119}\text{Sn}$ CP-CPMG	10	4	1024	-	-	-	6 ^d	1.1
S8C (MW on)	$^1\text{H} \rightarrow ^{119}\text{Sn}$ CP-CPMG	10	4	512	-	-	-	6 ^d	0.57

S8C (MW off)	$^1\text{H} \rightarrow ^{119}\text{Sn}$ CP-CPMG	10	4	1088	-	-	-	6^d	1.2
S8D (MW on)	$^1\text{H} \rightarrow ^{119}\text{Sn}$ CP-CPMG	10	4	64	-	-	-	6^d	0.07
S8D (MW off)	$^1\text{H} \rightarrow ^{119}\text{Sn}$ CP-CPMG	10	4	784	-	-	-	6^d	0.87
S8E (MW on)	$^1\text{H} \rightarrow ^{119}\text{Sn}$ CP-CPMG	10	2	1024	-	-	-	6^d	0.57
S8E (MW off)	$^1\text{H} \rightarrow ^{119}\text{Sn}$ CP-CPMG	10	2	2048	-	-	-	6^d	1.1
S9A	$2\text{D } ^1\text{H} \rightarrow ^{13}\text{C}$ CP-HETCOR	10	2	4	64	256	8.192	1^d	0.57
S9B	$2\text{D } ^1\text{H} \rightarrow ^{119}\text{Sn}$ CP-HETCOR	10	15.9	32	64	100	3.2	6^d	14.1
S12	$2\text{D } ^{119}\text{Sn}$ aMAT	10	10.4	42	3.4	136	0.232	2^d	16.5
S13	$2\text{D } ^{119}\text{Sn}$ aMAT	10	4	112	3.6	136	0.245	6^d	16.9
S14	$2\text{D } ^{119}\text{Sn}$ aMAT	10	4	112	3.6	138	0.248	6^d	17.2
S15	$2\text{D } ^{119}\text{Sn}$ aMAT	10	2	280	3.4	104	0.177	6^d	16.2

^a*J*-evolution duration. ^bQuantitative recycle delay ($\geq 5 \times T_1$). ^cRecycle delay *ca.* $4\text{-}5 \times T_1$. ^dCP contact time. ^eNumber of scans for each transmitter offset. The total number of ^{119}Sn offsets used to construct the VOCS spectrum is given in the parenthesis. ^fTotal experimental time for all ^{119}Sn transmitter offsets.

REVIEWERS' COMMENTS

Reviewer #1 (Remarks to the Author):

In my assessment, the authors have satisfactorily addressed the questions and comments I raised, making appropriate updates to their manuscript. Consequently, I believe it is suitable for acceptance and publication.

Reviewer #2 (Remarks to the Author):

The authors have addressed the comments of both reviewers with satisfactory changes and thus the manuscript is suitable for publication in its new form.